# Tunable Quantum Transport in Flux-Driven Designer Rings: Role of Hopping Dimerization, Electron Filling, and Phase Architecture

Souvik Roy [*1, 2, †] and Ranjini Bhattacharya[3, ‡]

[1]*School of Physical Sciences, National Institute of Science Education and Research, Jatni 752050, India*
[2]*Homi Bhabha National Institute, Training School Complex, Anushaktinagar, Mumbai 400094, India*
[3]*Department of Condensed Matter and Materials Physics,*
*S. N. Bose National Centre for Basic Sciences, JD-Block, Sector III, Salt Lake, Kolkata 700098, India*

We explore quantum transport in a one-dimensional Su–Schrieffer–Heeger (SSH) ring threaded by an Aharonov–Bohm flux, incorporating a quasiperiodic site potential based on a generalized cosine modulation with both primary and secondary components. Focusing on spinful, non-interacting electrons, we analyze persistent current behavior across a broad range of filling factors, Aubry–André–Harper (AAH) phase shifts, and hopping dimerizations. Our results uncover a rich interplay of localization–delocalization transitions, driven by the secondary modulation strength, leading to highly tunable and non-monotonic transport signatures. Persistent current is markedly enhanced near the symmetric hopping regime, where quantum interference dominates. In spin-imbalanced configurations, we identify regimes exhibiting pure spin current without net charge flow, suggesting potential for dissipationless spintronic functionalities. To the best of our knowledge, this is the first systematic study to examine quantum transport in such a flux-threaded SSH ring with dual-component quasiperiodic modulations, unifying re-entrant localization, spin-current generation, current stabilization, and phase-selective control. The inverse and normalized participation ratios, together with the state-resolved current, reveal the interplay between eigenstate localization and transport, providing insight into the microscopic mechanisms underlying current suppression and enhancement. For completeness, we also present the system-size dependence of the current. These findings position the quasiperiodic SSH ring as a versatile platform for realizing controllable quantum interference and spin-resolved transport in low-dimensional systems.

## I. INTRODUCTION

The interplay between disorder and quantum interference has long been central to our understanding of transport in low-dimensional systems. Since Anderson's seminal work [1,2], it is now a cornerstone of condensed matter physics that uncorrelated disorder in one-dimensional (1D) systems leads to the exponential localization of all eigenstates, effectively suppressing electronic diffusion and driving the system toward an insulating phase even at infinitesimal disorder strengths. This universal localization behavior in 1D random systems, while robust, was later shown to admit remarkable exceptions. The emergence of such quasiperiodic models most notably the Aubry-André-Harper (AAH) model[3–5] demonstrated that correlations in the potential landscape can profoundly modify localization properties, enabling controllable transitions between extended and localized states[6–16]. These findings have opened new avenues for engineering quantum transport through structured disorder, and continue to motivate exploration into tunable, non-random potentials in low-dimensional quantum materials.

A paradigmatic breakthrough in the study of deterministic localization is embodied in the AAH model[17–26], where the on-site potential follows a quasiperiodic modulation of the form $\epsilon_i = \lambda \cos(2\pi b i + \phi)$, with $\lambda$ controlling the modulation strength, $b$ an irrational incommensurate frequency, and $\phi$ a tunable phase. In contrast to Anderson-type disorder, the AAH model admits an exact self-duality that yields a sharp localization transition[27–36]: eigenstates remain extended for $\lambda < 2t$ and become exponentially localized beyond this critical value. This model has garnered considerable attention for its rich spectral features and experimental relevance across diverse platforms, including photonic lattices, ultracold atoms in bichromatic optical lattices, and engineered quasicrystalline materials. Recent theoretical developments have extended this framework by introducing spatially nonuniform or multi-component modulations within a unit cell, motivated by the growing interest in systems where internal lattice structure coexists with quasiperiodic order. These generalized AAH-type models break inversion symmetry, support intricate interference effects, and often exhibit nontrivial topological and transport characteristics. In particular, modulations that distinguish inequivalent sites within a dimer or superlattice structure offer new avenues for tuning localization behavior and controlling quantum coherence in engineered 1D systems.

An especially powerful probe of quantum coherence and spectral character in low-dimensional systems is the study of persistent currents in mesoscopic rings threaded by an Aharonov-Bohm (AB) flux[37–39]. These equilibrium currents, arising from the coherent circulation of electrons in a closed loop, are profoundly sensitive to the nature of the underlying eigenstates. In disorder-free systems, the current exhibits periodic oscillations with the applied flux, reflecting the underlying quantum interference around the ring. However, in the presence of disorder or quasiperiodicity, the amplitude and flux dependence of the current encode rich information about local-

───────
*Corresponding author: souvikroy138@gmail.com

ization, delocalization, and the degree of phase coherence in the system. Persistent currents have thus emerged as a valuable diagnostic tool both theoretically and experimentally[40–44] for characterizing transport regimes in mesoscopic systems[45–52]. A large body of work has leveraged these currents to explore the interplay between magnetic flux, electronic structure, and disorder-induced localization[53–56], making ring geometries an ideal setting for probing quantum interference phenomena in both ordered and aperiodic environments[57–62]. In rings with random or Aubry–André–Harper (AAH) site-energy modulation, the persistent current decreases monotonically with disorder[63] and vanishes in the strong limit, with next-nearest-neighbor hopping giving only minor enhancement. In contrast, nearest-neighbor hopping with unit-cell modulation yields a nonmonotonic response, showing a disorder-induced peak at half-filling[64,65], which naturally raises the question of realizing multiple peaks. The generalized AAH (GAAH)[66–69] model extends the conventional Aubry–André–Harper framework by introducing an additional tunable parameter, e.g., $\epsilon_n = \lambda \cos(2\pi b n + \phi)/[1 - \alpha \cos(2\pi b n + \phi)]$, allowing independent control of the numerator ($\lambda$) and denominator ($\alpha$). This provides two degrees of freedom to manipulate the quasiperiodic modulation, enabling the realization of mobility edges, nontrivial phase diagrams, and richer localization phenomena than in the original AAH model, making it a versatile platform for studying quasiperiodic transport and related effects. Although numerous studies have investigated localization phenomena in systems with GAAH potentials[70–74], *the behavior of the persistent current in such settings has, to the best of our knowledge, not been explored in the existing literature.* With this motivation, we investigate transport properties under non-interacting tight-binding framework based on the Su-Schrieffer-Heeger (SSH) lattice, incorporating GAAH unit-cell modulation in a one-dimensional ring geometry.

Our setup incorporates alternating nearest-neighbor hopping amplitudes along the chain and is configured in a ring geometry threaded by an Aharonov-Bohm (AB) magnetic flux, allowing access to persistent current measurements. *To introduce controlled aperiodicity, we impose a quasiperiodic onsite potential that acts asymmetrically on the two sublattices (A and B) within each unit cell. These potentials follow cosine modulations with distinct phase offsets, generating a spatially intricate potential landscape that breaks inversion symmetry at the microscopic level.* This asymmetric modulation framework enriches the spectral and transport properties of the SSH ring in profound ways. The combined influence of dimerization, sublattice-specific quasiperiodicity, and magnetic flux creates an ideal platform for investigating re-entrant localization-delocalization transitions, interference-driven spectral reshaping, and spin-dependent transport phenomena. The model offers tunability through hopping ratios, potential strengths, and magnetic phase, making it a fertile testbed for probing anomalous transport behaviors and flux-sensitive spectral responses in low-dimensional quantum systems.

Through extensive numerical simulations, we uncover a rich interplay between quasiperiodicity, lattice dimerization, and magnetic flux in shaping the transport properties of the SSH ring. *Our analysis demonstrates that the persistent charge and spin currents are highly sensitive to variations in the secondary potential strength $\lambda_2$, the hopping asymmetry ratio $t_1/t_2$, the internal phase configuration $(\phi_1, \phi_2)$, and the electronic filling factor. Remarkably, we observe multiple re-entrant transitions in charge transport as $\lambda_2$ is varied, signaling alternating regimes of localization and delocalization a behavior not typically expected in quasiperiodic systems. The charge current attains its maximum near the symmetric hopping regime $(t_1 = t_2)$, where interference between dimerization and potential modulation leads to enhanced coherence. The sublattice phase difference crucially modulates the transport landscape, underscoring the impact of internal unit-cell asymmetry. Furthermore, we find that quarter-filled systems display stronger transport signatures and more pronounced re-entrant features compared to half-filling, which tends to favor localization. Introducing spin imbalance opens pathways for generating pure spin currents even in the absence of net charge flow, an effect most prominent near the symmetric point. This unveils a dissipationless spin transport channel governed by quantum interference.* Complementing the current analysis, spectral diagnostics based on the inverse participation ratio (IPR) and normalized-participation ratio (NPR), along with their averaged counterparts, reveal a strong correlation between eigenstate delocalization and transport enhancement. Intriguingly, in certain parameter regimes, increasing the quasiperiodic strength $\lambda_2$ counterintuitively facilitates delocalization, highlighting a re-entrant transport behavior beyond standard expectations.

The remainder of this work is structured to guide the reader through the physical formulation, computational exploration, and broader implications of our results. In Sec. II A, we construct the tight-binding Hamiltonian for the SSH ring geometry, incorporating dimerized hopping, a two-sublattice quasiperiodic modulation, and a threaded Aharonov-Bohm flux. In Sec. II B, we compute the ground state energy and evaluate the associated persistent current to capture the system's coherent transport response under flux. Section II C presents a detailed analysis of eigenstate localization using normalized participation ratio (NPR) and inverse participation ratio (IPR), revealing strong correlations between transport enhancement and spectral delocalization. The results, discussed extensively in Sec. III, uncover a rich interplay of hopping asymmetry, phase modulation, and spin imbalance, leading to re-entrant localization transitions, regimes of dissipationless spin transport, and experimental realization. Finally, Sec. IV provides a synthesis of our key findings and proposes avenues for realizing controllable quantum interference and spin-filtering phenomena in engineered low-dimensional aperiodic systems. Technical details and supplementary results are presented in the Appendix.

## II. THEORETICAL FRAMEWORK

### A. Model Description

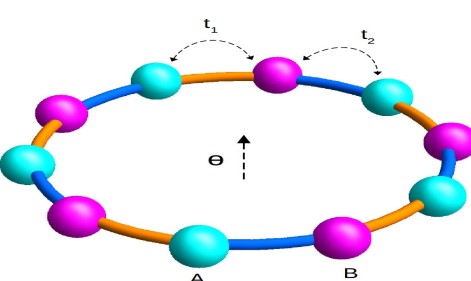

FIG. 1: Schematic representation of a quasiperiodic Su-Schrieffer-Heeger (SSH) ring threaded by an external Aharonov-Bohm flux, where the on-site energies are modulated via a generalized Aubry-André-Harper (AAH) potential. This engineered architecture showcases the synergistic interplay between flux-induced quantum interference and internal structural modulation, revealing a tunable and robust framework for amplifying persistent currents.

We consider a one-dimensional system of spin-$\frac{1}{2}$ fermions confined to a mesoscopic ring composed of $N$ discrete lattice sites. This platform serves as a fertile ground to explore the rich interplay among structural quasiperiodicity, and quantum coherence. The system is engineered to host a generalized AAH-type quasiperiodic modulation in its onsite potential landscape, and is further threaded by a tunable Aharonov-Bohm magnetic flux $\phi$, which introduces controlled quantum interference effects across the ring (see Fig. 1).

Our model extends the canonical Su-Schrieffer-Heeger (SSH) tight-binding framework in multiple directions: it incorporates spin-resolved dynamics, spatially inhomogeneous onsite energies following a quasiperiodic unit cell sequence, that mediate correlation-driven phenomena. This confluence of SSH dimerization, AAH aperiodicity, and magnetic flux makes the system an ideal setting for probing the emergence of exotic transport regimes, reentrant localization-delocalization behavior, and tunable spin-charge separation in low-dimensional correlated systems.

The tight-binding Hamiltonian governing the system is given by

$$\mathcal{H} = \sum_{j \in \text{even},\sigma} \left( t_1 \, e^{2\pi i\theta/N\theta_0} \, c_{j,\sigma}^\dagger c_{j+1,\sigma} + \text{H.c.} \right)$$
$$+ \sum_{j \in \text{odd},\sigma} \left( t_2 \, e^{2\pi i\theta/N\theta_0} \, c_{j,\sigma}^\dagger c_{j+1,\sigma} + \text{H.c.} \right)$$
$$+ \sum_{k=A,B} \sum_{j,\sigma} \epsilon_{k,j} \, n_{j,\sigma}, \quad (1)$$

where $c_{j,\sigma}^\dagger$ ($c_{j,\sigma}$) creates (annihilates) a fermion with spin $\sigma = \uparrow, \downarrow$ at site $j$, and $n_{j,\sigma} = c_{j,\sigma}^\dagger c_{j,\sigma}$ is the corresponding number operator. The hopping amplitudes $t_1$

and $t_2$ alternate along the lattice, capturing the characteristic dimerization of the Su-Schrieffer-Heeger (SSH) model. The complex phase factor $e^{2\pi i\theta/N\theta_0}$ stems from the Peierls substitution and encodes the effect of an external Aharonov-Bohm magnetic flux $\theta$ (in the unit of $\theta_0$) threading the ring, introducing flux-dependent interference that modulates coherent transport.

Each unit cell contains two sublattice sites, labeled $A$ and $B$, whose onsite potentials are modulated quasiperiodically. The onsite energy at site $j$ belonging to sublattice $k = A, B$ is given by the expressions $\epsilon_{A,j} = \lambda_1 \cos(2\pi bj)/\left(1 + \lambda_2 \cos(2\pi bj + \phi_1) + \eta\right)$ and $\epsilon_{B,j} = \lambda_1 \cos(2\pi bj)/\left(1 + \lambda_2 \cos(2\pi bj + \phi_2) + \eta\right)$, where $\lambda_1$ and $\lambda_2$ define the amplitudes of the primary and secondary quasiperiodic modulations, respectively. The irrational parameter $b$ ensures an incommensurate modulation, while $\phi_1$ and $\phi_2$ are phase offsets that asymmetrically shift the potential landscapes of the two sublattices. A small regularization parameter $\eta$ prevents divergence in the potential and stabilizes the modulation. *We define the non-staggered phase configuration by setting $\phi_1 = 0$ and $\phi_2 = 0$, while the staggered phase corresponds to $\phi_1 = 0$, $\phi_2 = \pi$. Intermediate phase modulation refers to configurations with fixed $\phi_1 = 0$ and arbitrary finite $\phi_2 \neq 0, \pi$, enabling a continuous tuning between symmetric and antisymmetric phase patterns.* This asymmetric dual-phase modulation creates a spatially intricate potential landscape, which, when combined with SSH dimerization and magnetic flux, enables a rich spectrum of localization phenomena and tunable charge and spin transport in mesoscopic settings.

This model offers a compelling platform to investigate the evolution of topological and transport properties under the intertwined influence of quasiperiodic order, magnetic flux, and electron filling. The flux-dependent hopping terms naturally encode Aharonov-Bohm-type quantum interference effects, enabling precise control over coherent transport through external magnetic fields. Simultaneously, the AAH-modulated onsite potential introduces aperiodicity, fostering a unique landscape that supports both critical and localized eigenstates, hallmarks of nontrivial quantum phases in quasiperiodic systems.

### B. Ground-State Energy and Flux-Driven Persistent Currents

To unveil the coherent transport characteristics of the system, we begin by evaluating the ground-state energy $E_0$ at zero temperature. It is obtained by summing over all occupied single-particle energy levels for both spin channels:

$$E_0 = \sum_{j=1}^{n_\uparrow} E_{j,\uparrow} + \sum_{j=1}^{n_\downarrow} E_{j,\downarrow}, \quad (2)$$

where $E_{j,\sigma}$ denotes the $j^{\text{th}}$ eigenenergy for spin $\sigma \in \{\uparrow, \downarrow\}$, and $n_\sigma$ corresponds to the number of filled states for each spin sector, determined by the total filling and spin imbalance.

A central observable in mesoscopic rings is the persistent current, which emerges from the sensitivity of quantum states to magnetic flux penetrating the ring. This current serves as a direct fingerprint of quantum coherence and topological boundary conditions. The total persistent charge current is extracted from the flux derivative of the ground-state energy:

$$I = -c\frac{\partial E_0}{\partial \theta}, \tag{3}$$

where $\theta$ is the flux (with $\theta_0$ the flux quantum), and $c$ is the speed of light. The periodicity of $I$ as a function of $\theta$ reflects the Aharonov-Bohm interference encoded in the system's energy spectrum.

To further disentangle the spin-dependent transport behavior, we resolve the contributions from individual spin channels. The spin-resolved persistent currents are defined as:

$$I_\uparrow = -c\frac{\partial E_\uparrow}{\partial \theta}, \quad I_\downarrow = -c\frac{\partial E_\downarrow}{\partial \theta}, \tag{4}$$

with $E_\uparrow$ and $E_\downarrow$ denoting the ground-state energy contributions from up-spin and down-spin electrons, respectively. These allow for the identification of spin-polarized transport responses. The net circular spin current, a hallmark of broken spin symmetry or interaction-induced effects, is given by:

$$I_{\text{spin}} = I_\uparrow - I_\downarrow, \tag{5}$$

while the total charge current follows from their sum:

$$I_{\text{charge}} = I_\uparrow + I_\downarrow. \tag{6}$$

Unless otherwise stated, $I$ refers to the total charge current, while spin-resolved quantities are explicitly denoted.

To study transport at the level of individual eigenstates, we calculate the state-resolved current along the lattice. The current operator is defined as

$$\hat{\mathcal{J}} = \frac{e\,\hat{\mathcal{V}}}{N},$$

where $\hat{\mathcal{V}}$ is the velocity operator and $N$ denotes the system length. In quantum mechanics, the velocity operator can be expressed as the time derivative of the position operator through the commutator with the Hamiltonian:

$$\hat{\mathcal{V}} = \frac{d\hat{\mathcal{X}}}{dt} = \frac{1}{i\hbar}[\hat{\mathcal{X}}, H].$$

Here, the position operator is given by

$$\hat{\mathcal{X}} = \sum_{n,\sigma} n\, d_{n,\sigma}^\dagger d_{n,\sigma},$$

where $d_{n,\sigma}^\dagger$ ($d_{n,\sigma}$) creates (annihilates) an electron of spin $\sigma$ at site $n$. Evaluating the commutator with the dimer-ized Hamiltonian leads to the spin-resolved current operator

$$\hat{\mathcal{J}}_\sigma = \frac{2\pi e}{i\hbar N}\left[ \sum_{n\in\text{even}} \left(t_1 e^{-2\pi i\theta/N\theta_0} d_{n+1,\sigma}^\dagger d_{n,\sigma} - h.c\right) \right.$$
$$\left. + \sum_{n\in\text{odd}} \left(t_2 e^{-2\pi i\theta/N\theta_0} d_{n+1,\sigma}^\dagger d_{n,\sigma} - h.c.\right) \right], \tag{7}$$

with $\theta$ representing the magnetic flux and $t_1$, $t_2$ the alternating hopping amplitudes. The current corresponding to an eigenstate $|\chi_m\rangle$ is obtained as

$$\langle \chi_m|\hat{\mathcal{J}}_\sigma|\chi_m\rangle.$$

The eigenstate can be expanded in the Wannier basis as

$$|\chi_m\rangle = \sum_{n,\sigma} b_{n,\sigma}^{(m)}|n,\sigma\rangle, \tag{8}$$

where $b_{n,\sigma}^{(m)}$ denote the complex amplitudes at site $n$, spin $\sigma \in \{\uparrow,\downarrow\}$, and eigenstate index $m$. Substituting this expansion into the expectation value of the current operator gives

$$\mathcal{J}_\sigma^{(m)} = \frac{2\pi i e}{N\hbar} \sum_{\sigma=\uparrow,\downarrow}\left[ \sum_{n\text{ odd}} t_1\left(e^{2\pi i\theta/N\theta_0} b_{n,\sigma}^{(m)*} b_{n+1,\sigma}^{(m)} - \text{h.c.}\right) \right.$$
$$\left. + \sum_{n\text{ even}} t_2\left(e^{2\pi i\theta/N\theta_0} b_{n,\sigma}^{(m)*} b_{n+1,\sigma}^{(m)} - \text{h.c.}\right) \right]. \tag{9}$$

For simplicity, we denote the eigenstate-resolved current $\mathcal{J}_\sigma^{(m)}$ as $J_{m,\sigma}$ throughout our results. This formalism provides a way to compute the contributions of individual eigenstates to the total persistent current in dimer-ized lattices threaded by magnetic flux, revealing detailed insights into eigenstate-dependent transport.

### C. Probing Eigenstate Localization via Participation Ratios

To characterize the spatial nature of the eigenstates, ranging from extended to localized to critical, we employ two diagnostic tools: the inverse participation ratio (IPR) and the normalized participation ratio (NPR). These quantities are computed from the normalized single-particle wavefunctions $\psi^n = \{\psi_p^n\}$ of the $n$th eigenstate, where $p$ labels the lattice sites.

The IPR, defined as

$$\text{IPR}_n = \sum_p |\psi_p^n|^4, \tag{10}$$

measures the degree of localization of an individual eigenstate. For a completely delocalized state, $\text{IPR}_n \sim 1/N$, whereas it approaches unity for a state localized on a single site.

Complementarily, the NPR is given by

$$\text{NPR}_n = \left( N \sum_p |\psi_p^n|^4 \right)^{-1}, \qquad (11)$$

which scales toward unity for extended states and vanishes for localized ones. This dual characterization enables a robust identification of mobility edges and critical regimes.

To capture the global nature of the eigenstate landscape across the full spectrum, we compute the average NPR and average IPR over all $N$ single-particle eigenstates:

$$\langle \text{NPR} \rangle \text{ or } \langle \text{IPR} \rangle = \frac{1}{N} \sum_{n=1}^{N} \text{NPR}_n \text{ or } \frac{1}{N} \sum_{n=1}^{N} \text{IPR}_n. \quad (12)$$

This averaged measure provides a powerful diagnostic to map out localization transitions and compare different regimes of disorder, interaction strength, and magnetic flux. Together with the persistent current analysis, it forms a comprehensive toolkit to study the intricate interplay of coherence, quasi-periodicity, and correlation in this versatile platform.

## III.   NUMERICAL RESULTS AND DISCUSSIONS

In this section, we systematically uncover the intricate transport characteristics of spinful electrons in a quasiperiodically modulated SSH ring, focusing on how the combined effects of quasiperiodic potential strength, hopping dimerization, and spin imbalance govern the behavior of persistent currents. To dissect the interplay between intra-cell and inter-cell tunneling, we analyze three distinct dimerization regimes,namely, the uniform hopping case ($t_1 = t_2$), the intra-cell suppressed case ($t_1 < t_2$), and the inter-cell suppressed case ($t_1 > t_2$). *Throughout this work, we anchor the primary AAH phase at $\phi_1 = 0$ and systematically investigate three distinct designer flux-phase architectures stemming from the secondary modulation: Model 1 embodies the non-staggered regime with $\phi_2 = 0$, Model 2 implements a staggered phase $\phi_2 = \pi$, while Model 3 captures the intermediate scenario via a tunable phase $\phi_2 = \pi/2$, offering a bridge between the two extremes. To uncover spin-resolved transport characteristics, we introduce tailored spin-imbalance configurations, type-1, featuring an even up-spin and odd down-spin population, and type-2, with even populations in both spin sectors, enabling a fine-grained analysis of spin-polarized current dynamics in quasiperiodic designer rings.* System sizes are provided in the captions of the plots, with disorder and hopping parameters specified in the respective figures. To characterize the localization-delocalization landscape underlying the observed current profiles, we further evaluate disorder-averaged inverse participation ratio (IPR), normalized participation ratio (NPR), and eigenstate-resolved NPR spectra. Unless stated otherwise, we take $\lambda_1 = 1.5$ throughout. All energy-related parameters and disorder modulation strengths are expressed in units of electronvolts (eV), while the persistent current is measured in microamperes ($\mu$A).

### A.   Flux-Tuned NPR-Resolved Energy Spectrum

Figure 2 encapsulates the flux-resolved single-particle spectra, color-coded by the normalized participation ratio (NPR), for representative configurations of hopping asymmetry and AAH phase offsets. Each row delineates a distinct dimerization pattern in the intra-chain hopping amplitudes namely, $t_1 = t_2$ (top), $t_1 > t_2$ (middle), and $t_1 < t_2$ (bottom) while the columns span different quasiperiodic phase combinations ($\phi_1, \phi_2$), with $\phi_1$ held fixed and $\phi_2$ varied across 0, $\pi$, and $\pi/2$. For the uniform hopping scenario ($t_1 = t_2$), the spectrum under $\phi_2 = 0$ displays densely packed, highly dispersive energy bands with elevated NPR values, indicating largely extended states. This delocalized nature, in conjunction with narrow level spacing, favors robust flux sensitivity and enhances persistent current. As $\phi_2$ is tuned to $\pi$, while the NPR remains substantial, a central spectral gap becomes apparent and the curvature of the bands near zero energy flattens, indicating a diminished group velocity and thus reduced current. The intermediate case of $\phi_2 = \pi/2$ retains a continuous spectrum with moderate NPR values and visibly steeper energy bands near the zero energy level, supporting relatively stronger flux-driven currents (near half filling) than the gapped $\phi_2 = \pi$ configuration. Interestingly, for $\phi_2 = \pi/2$, we observe the emergence of nearly flat energy levels at the spectral edges, accompanied by a pronounced suppression in the normalized participation ratio (NPR), signaling strong localization at the edges of the spectrum. When hopping asymmetry is introduced ($t_1 > t_2$, middle row), a marked band separation emerges near zero energy for all $\phi_2$, signaling an increase in localization and a general suppression of current. Nevertheless, among these, $\phi_2 = 0$ still supports the most extended eigenstates and hence the highest current, reaffirming the sensitivity of transport to the interplay between phase and hopping profile. In the $t_1 < t_2$ regime (bottom row), the spectrum exhibits the most pronounced subband fragmentation and lowest NPR values, particularly for $\phi_2 = \pi/2$, where a significant gap opens near zero energy. This strongly localized character renders the system effectively insulating, with minimal spectral response to flux variation. Consequently, at half-filling, the persistent current is substantially suppressed across all phase configurations, highlighting the extreme sensitivity of transport to quasiperiodic modulation under hopping inversion. In contrast, at quarter-filling, the relevant energy bands remain sufficiently close in energy across all configurations, preserving notable band curvature and elevated NPR values. This indicates the presence of more extended states and enhanced flux sensitivity, thereby supporting larger persistent currents for all combinations of hopping asymmetry and phase settings. Collectively, these results illuminate the delicate

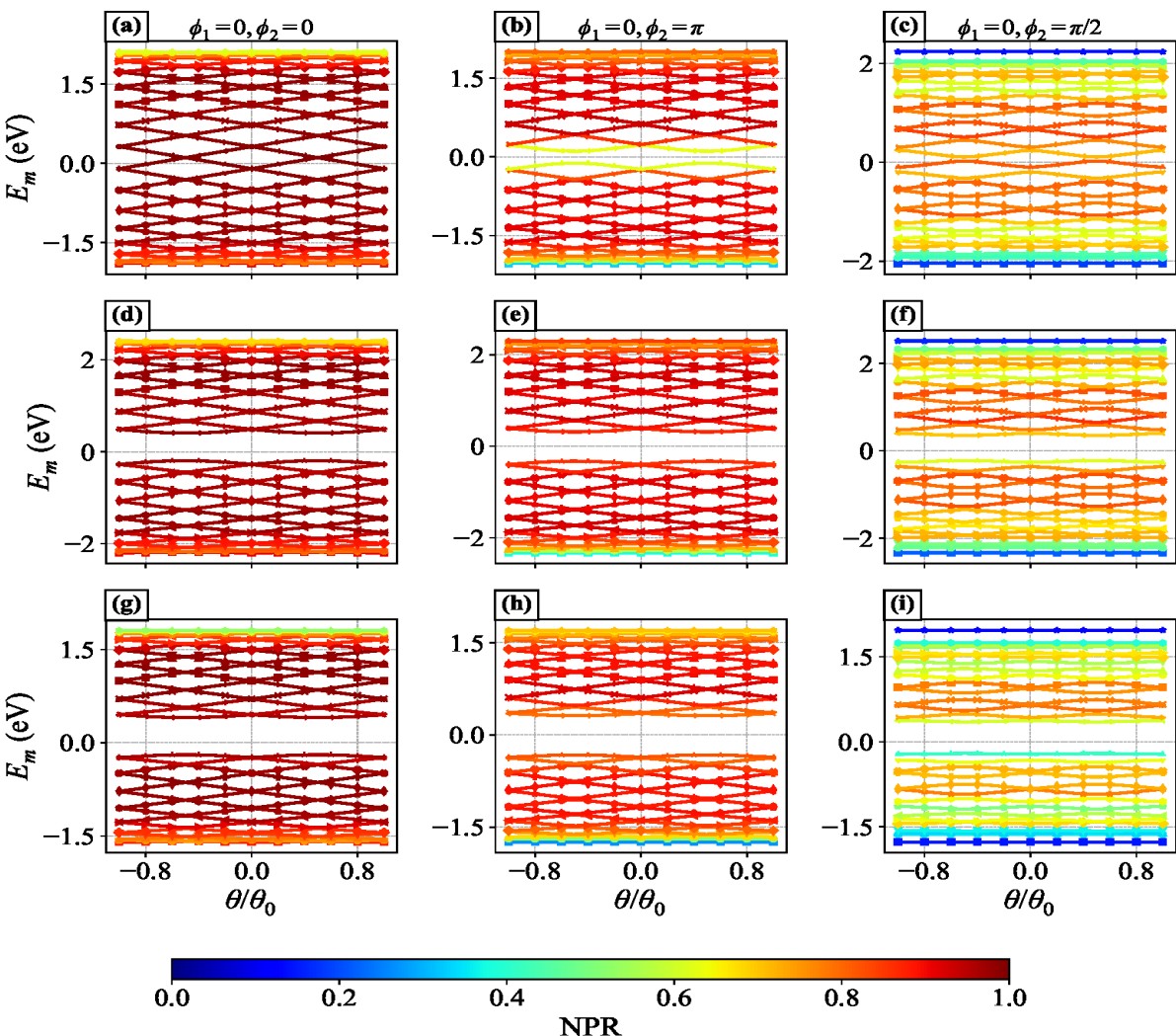

FIG. 2: (Color online). Flux-dependent variation of NPR resolved energy levels, revealing the interplay between magnetic flux and quasiperiodic modulation in engineered SSH ring structures. The first row [panels (a), (b), and (c)] corresponds to the uniform hopping case ($t_1 = t_2 = 1$), with three distinct Aubry phase configurations: $(\phi_1, \phi_2) = (0,0)$, $(0,\pi)$, and $(0,\pi/2)$, respectively. The second row [panels (d), (e), and (f)] explores the impact of hopping dimerization with $t_1(1.3) > t_2(1.0)$ under such phase configurations. The third row [panels (g), (h), and (i)] illustrates the complementary regime with $t_1(0.7) < t_2(1.0)$, again employing the same sequence of phase choices. Since the energy eigenvalues for up and down spins are identical across the parameter space, we present spin-resolved observables using their average spectrum. In this case, $\lambda_1 = 1.5$, $\lambda_2 = 15$, with a total of 30 sites.

and tunable interdependence of quasiperiodic phase disorder and hopping asymmetry on quantum transport: by modulating $(t_1/t_2, \phi_2)$, one can strategically manipulate spectral continuity, localization strength, and flux sensitivity, offering a versatile mechanism for controlling persistent currents in engineered aperiodic systems.

## B. Tuning Ground State Energy and Current via Flux

In Fig. 3, we present a comprehensive study of the ground-state energy and its flux-induced derivative, the persistent current across various relative phase configurations and disorder strengths, all evaluated at quarter-filling to ensure densely energy states and robust trans-

port. The first row displays the evolution of the ground-state energy as a function of magnetic flux, while the second row reports the corresponding current profiles. Each panel features three distinct phase arrangements ($\phi_2 = 0$, $\pi$, $\pi/2$) between the two atoms in a unit cell (showing with red, green, and blue curves), and the columns represent increasing strengths of the secondary quasiperiodic potential $\lambda_2 = 15$, 22, and 30, respectively. For weak disorder ($\lambda_2 = 15$), the red and green configurations show strong flux sensitivity with discernible curvature in the energy spectrum, signifying enhanced coherence and enabling larger persistent currents. The blue configuration, on the other hand, appears nearly flux-insensitive its energy profile is substantially flatter implying suppressed transport. This qualitative behavior is quantitatively confirmed in Fig. 3(d), where the currents showing

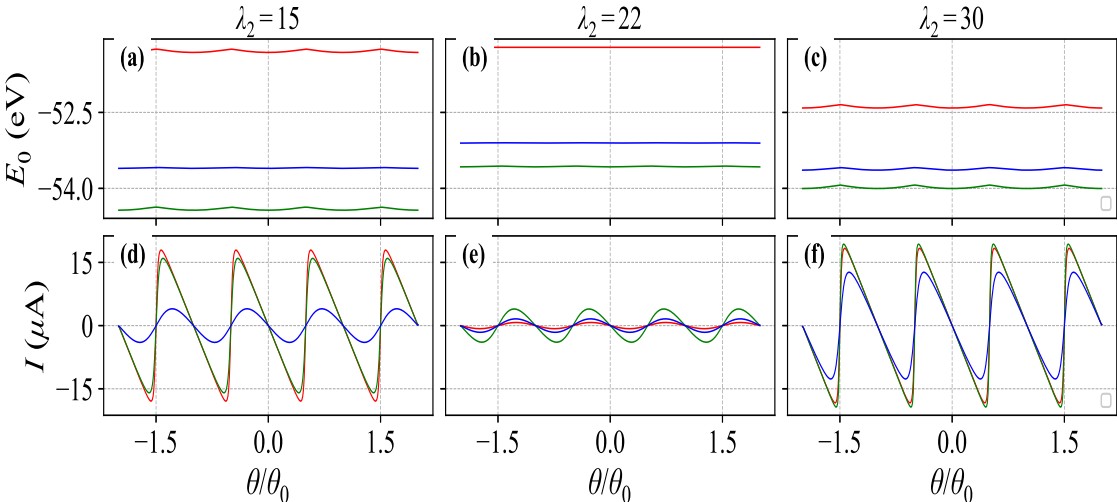

FIG. 3: (Color online). Ground-state energy and the corresponding persistent current are shown as functions of the magnetic flux $\theta$ for the case $t_1 = t_2 = 1$ at quarter filling with $\lambda_1$ fixed at 1.5, highlighting the impact of quasiperiodic modulation and flux phase tuning on quantum transport. The top row [panels (a), (b), and (c)] illustrates the flux dependence of the ground state energy for increasing disorder strengths, $\lambda_2 = 15$, 22, and 30, respectively. Each panel contrasts three distinct Aubry phase configurations $\phi_2 = 0$, $\pi$, and $\pi/2$, shown in red, green, and blue curves, respectively, to capture phase-sensitive interference effects. The bottom row [panels (d), (e), and (f)] displays the corresponding variation of persistent current under the same conditions, demonstrating how increasing disorder and phase modulation reshape the flux-periodic current. Here we consider a lattice of total $N = 60$ sites.

with red and green colors manifest prominent amplitudes, while the blue one remains nearly quenched.

As the quasiperiodic potential strength increases to $\lambda_2 = 22$, the red curve flattens out significantly [Fig. 3(b)], indicating localization-induced suppression of current, consistent with the diminished response seen in Fig. 3(e). However, the green and blue curves exhibit slight variations with flux, indicating the presence of a finite current, which remains notably higher compared to that of the red curve. Intriguingly, at even stronger disorder ($\lambda_2 = 30$), a partial revival of flux sensitivity is observed in the red and green energy spectra [Fig. 3(c)], leading to a corresponding resurgence in current amplitude [Fig. 3(f)]. This nonmonotonic behavior underscores a subtle and rich interplay between quasiperiodic disorder and AAH phase. Notably, the relative phase between the atoms of unit cell emerges as a powerful tuning knob acting as a flux-sensitive switch that modulates the degree of coherence and transport. Moreover, the efficacy of this phase-control mechanism is strongly influenced by the strength of the underlying secondary aperiodic modulation. These findings point to a compelling avenue for engineering tunable mesoscopic currents via controlled phase manipulation in quasiperiodic correlated systems.

## C. Effect of Secondary Modulation $\lambda_2$ on Current at Fixed Flux

In Fig. 4, we investigate the behavior of the persistent current at a fixed magnetic flux ($\theta = 0.3\theta_0$) as a function of the quasiperiodic potential strength $\lambda_2$, for both half-filling (top row) and quarter-filling (bottom row). Each column captures a different correlation scenario between the hopping amplitudes: symmetric hopping ($t_1 = t_2$), and asymmetric cases with $t_1 > t_2$ and $t_1 < t_2$. Within each panel, the impact of relative phase $\phi_2$ between the chains is highlighted through red ($\phi_2 = 0$), green ($\phi_2 = \pi$), and blue ($\phi_2 = \pi/2$) curves, with $\phi_1$ fixed at zero. In the symmetric hopping case in the context of half-filling [Fig. 4(a)], the configuration $\phi_2 = 0$ stands out by producing the highest current magnitudes, punctuated by multiple peaks and dips that signal re-entrant behavior driven by competing localization and coherence. These oscillations point to a sensitive interference landscape that modulates transport resonance conditions. In contrast, $\phi_2 = \pi$ also yields moderate currents but shows late saturation of current with $\lambda_2$ than $\phi_0$ case, while $\phi_2 = \pi/2$ results in the most gradual current rise, reflecting weaker transport response and the current tends to saturate at higher values for increasing $\lambda_2$, surpassing the corresponding saturation levels observed in the other two cases. So interestingly, both $\phi_2 = 0$ and $\pi$ configurations reach a regime of current saturation at lower $\lambda_2$ than $\phi_2 = \pi/2$ case, indicating the emergence of transport plateaus that remain largely insensitive to further increases in disorder, marking a quasi-steady transport window even in the presence of strong aperiodicity.

As the hopping asymmetry is introduced, new transport phenomena emerge. For $t_1 > t_2$ [Fig. 4(b)], the $\phi_2 = 0$ configuration continues to dominate, featuring sharp current resonances that eventually fade with increasing $\lambda_2$, signaling a crossover from delocalized to localized behavior. For $t_1 < t_2$, a striking reversal in the current behavior is observed [Fig. 4(c)]. In this regime, the previously dominant $\phi_2 = 0$ and $\pi$ channels are al-

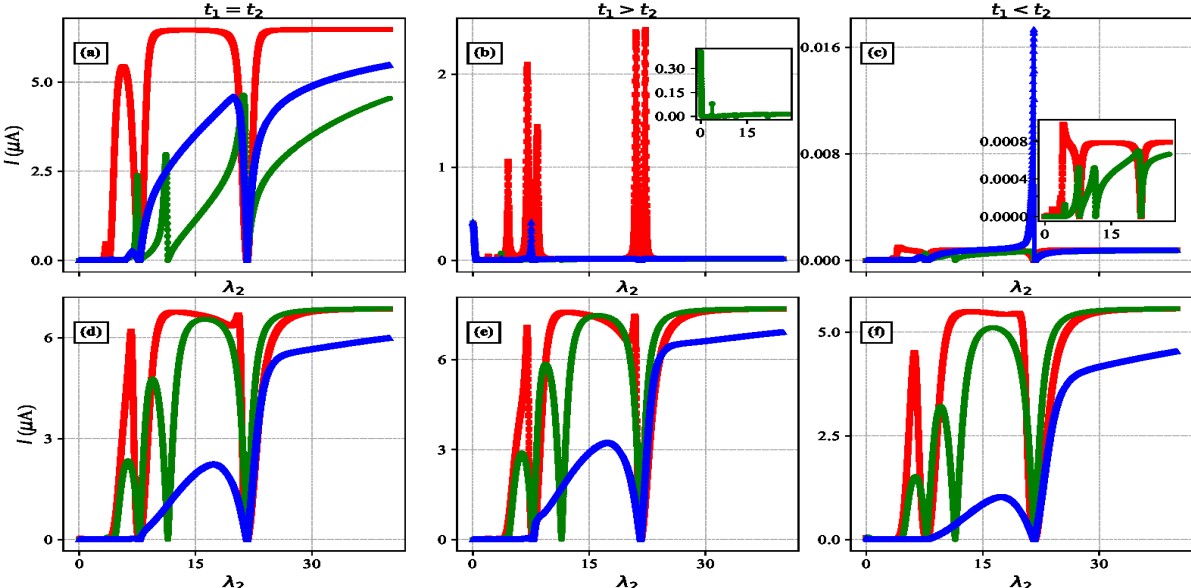

FIG. 4: (Color online). Disorder-tuned evolution of the persistent current in a quasiperiodic SSH ring threaded by a fixed Aharonov-Bohm flux of $0.3\theta_0$ for $\lambda_1 = 1.5$, showcasing how the interplay between hopping asymmetry and phase modulation governs transport properties. The first row [panels (a), (b), and (c)] corresponds to the half-filled case for three hopping configurations: uniform hopping ($t_1 = t_2 = 1$), dimerized with $t_1(1.3) > t_2(1.0)$, and $t_1(0.7) < t_2(1.0)$, respectively. The second row [panels (d), (e), and (f)] displays analogous results for the quarter-filled scenario. In each panel, the current is plotted as a function of disorder strength $\lambda_2$, with red, green, and blue curves denoting the phase values $\phi_2 = 0$, $\pi$, and $\pi/2$, respectively for $N = 60$.

most completely quenched, showing only negligible transport. This suppression arises because the smaller intracell hopping $t_1$ limits coherent motion within the unit cell, and for these phase choices, destructive interference between intra- and inter-unit-cell hoppings further reduces the current. In contrast, the $\phi_2 = \pi/2$ configuration sustains a significantly higher current, as the mixed cosine-sine modulation of the site potentials allows better phase matching between the hopping pathways, facilitating coherent transport. While the $\phi_2 = 0$ and $\pi$ channels still exhibit multiple peaks, dips, and re-entrant transitions, their overall current magnitude remains substantially lower than that of $\phi_2 = \pi/2$, highlighting the crucial role of the relative phase in controlling transport efficiency.

The situation markedly improves at quarter-filling [Figs. 4(d)–(f)], where across all hopping asymmetries, the current remains more resilient to increasing $\lambda_2$, reflecting the role of densely populated energy states exhibiting significant curvature with flux and associated with higher NPR values which preserving extended states. Notably, even under strong quasiperiodic modulation, several phase configurations exhibit current saturation behavior, demonstrating robustness against disorder. Among these, the $\phi_2 = \pi/2$ phase consistently shows delayed saturation and fewer resonant peaks, suggesting suppressed but long-lived transport channels. Another interesting observation is that at half-filling, the current follows the dominance ordering $I_{\phi_2=0} > I_{\phi_2=\pi/2} > I_{\phi_2=\pi}$, whereas at quarter-filling, the ordering reverses to $I_{\phi_2=0} > I_{\phi_2=\pi} > I_{\phi_2=\pi/2}$. This indicates that the stag-

gered and intermediate phase configurations exchange their roles as the filling is tuned from half to quarter. Collectively, these results showcase how tuning the interplay between hopping asymmetry, filling fraction, and unit cell phase configuration offers powerful control over transport characteristics paving the way for custom design of persistent current devices in quasiperiodic quantum architectures.

### D. Non-Staggered Phase: Current Variation with Disorder Strength and Hopping Dimerization

To further isolate the impact of phase modulation on transport, we consider a simplified variant of our original tight-binding model in which the secondary AAH phase remains uniform across both sublattice sites within each unit cell. By fixing $\phi_1 = \phi_2 = 0$, we effectively eliminate the site-dependent phase asymmetry while preserving the overall quasiperiodic structure of the potential. This configuration restores intra-cell translational symmetry but maintains the aperiodicity at the lattice scale, allowing us to disentangle the influence of local phase gradients from that of the global quasiperiodic modulation. Figures 5 and 6 present a comprehensive analysis of the resulting transport behavior under this phase-homogeneous setup, highlighting both charge and spin current responses for spin-balanced and spin-imbalanced scenarios. The findings serve as a critical reference point for assessing the role of inter-site phase contrast in modulating persistent current characteristics.

### 1. Charge Transport Response to Hopping Ratio and $\lambda_2$

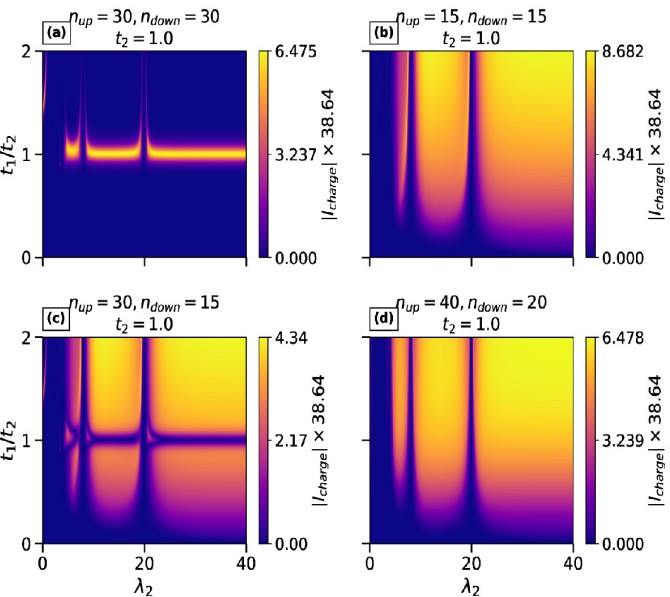

FIG. 5: (Color online). Behavior of charge current as a function of quasiperiodic disorder strength $\lambda_2$ and hopping asymmetry ratio $t_1/t_2$ at a fixed Aharonov-Bohm flux of $0.3\,\theta_0$ and $\phi_2 = 0$, illustrating the interplay between transport asymmetry and controlled disorder. Panel (a) depicts the half-filled regime, while panel (b) explores the quarter-filled case. Panels (c) and (d) highlight imbalanced spin populations, uncovering how spin-selective configurations lead to asymmetric current responses under identical flux and disorder settings.

Figure 5(a) presents the phase diagram of the charge current, $I_{\text{charge}}$, as a function of the hopping ratio $t_1/t_2$ and the quasiperiodic disorder strength $\lambda_2$ in the half-filled configuration. A pronounced peak in the current emerges around $t_1/t_2 \approx 1$, spanning a broad window of disorder values. This central maximum signals an optimal regime where balanced hopping enhances coherent charge transport despite moderate disorder. Deviations from this balance, i.e., when $t_1/t_2$ moves away from unity, result in a marked suppression of the current, pointing to the fragility of transport under asymmetry. Intriguingly, near the symmetric point $t_1 = t_2$, the system exhibits a rich structure of re-entrant behavior manifested as repeated high-to-low transitions in current with increasing $\lambda_2$. These oscillations reflect alternating regimes of constructive and destructive quantum interference, rooted in the subtle interplay between hopping asymmetry and quasiperiodic modulation. Such dynamical sensitivity highlights the potential for multi-level current switching via microscopic tuning within a specific parameter regime.

The quarter-filled counterpart, depicted in Fig. 5(b), reveals an even more compelling scenario. Here, $I_{\text{charge}}$ achieves significantly higher magnitudes across a broader swath of the $t_1/t_2$ axis. The re-entrant features become sharper and more frequent, persisting over a wider range of $\lambda_2$ values. This amplified response at lower fillings is

likely tied to the asymmetric occupation of energy bands and the emergence of resonant transmission conditions in sparser electronic environments, which, in turn, heightens the system's exposure to interference and disorder.

We next examine the charge current under spin-imbalanced conditions, where the number of spin-up and spin-down electrons differs. Figure 5(c) illustrates the phase diagram for a configuration in which spin-up (even) and spin-down (odd) populations are unequal. Remarkably, the charge current becomes significantly suppressed around $t_1/t_2 \approx 1$, forming a broad low-current plateau across a wide range of disorder strengths, strikingly opposite to the enhanced transport observed in the spin-balanced case. Furthermore, for fixed disorder strengths near $\lambda_2 \approx 10$ and 20, this suppression remains robust across the entire hopping ratio domain, signaling a global degradation of coherent charge transport due to spin imbalance.

A reversed spin configuration is explored in Fig. 5(d), where the down-spin electrons now occupy even sites. In this setting, the system regains much of its transport efficiency, with $I_{\text{charge}}$ attaining higher values over an extended disorder range. The current profile here closely mirrors that of the quarter-filled spin-balanced case (see Fig. 5(b)), suggesting that spin imbalance can either hinder or facilitate transport depending on the parity and spatial distribution of the electron population.

Together, these observations underscore the delicate and highly tunable nature of transport in quasiperiodic lattices with dimerized hopping. The emergence of re-entrant current features, transport maxima, and spin-sensitive suppression regimes reflects a complex competition among quasiperiodic modulation, hopping asymmetry, and spin configuration. These insights open up a promising pathway toward engineering controllable transport phenomena in low-dimensional quantum systems. Specifically, by tailoring the parameters $t_1/t_2$, $\lambda_2$, and the spin population imbalance, one can enable current modulation, with potential applications in quantum interference-based switching and phase-controlled nanoelectronic device architectures.

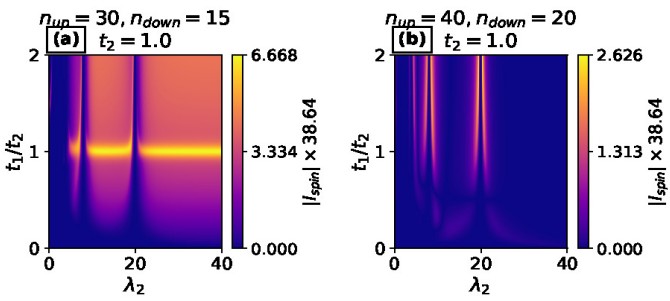

FIG. 6: (Color online). Spin current as a function of quasiperiodic disorder strength $\lambda_2$ and hopping ratio $t_1/t_2$, evaluated for fixed phase $\phi_2 = 0$ and $\theta = 0.3\,\theta_0$. Panels (a) and (b) explore regimes with unequal spin populations, revealing how the interplay between hopping asymmetry and correlated disorder gives rise to spin-selective transport.

### 2. Spin current modulation under disorder and spin imbalance

We now turn our attention to the behavior of spin transport in the presence of spin imbalance, focusing on how disorder and hopping asymmetry influence the spin current $I_{\mathrm{spin}}$. Figure 6(a) displays the phase diagram of $I_{\mathrm{spin}}$ as a function of the hopping ratio $t_1/t_2$ and the on-site quasiperiodic disorder strength $\lambda_2$, for a particular spin-imbalanced configuration (odd down spin). Strikingly, the spin current attains its peak magnitude in the vicinity of the symmetric hopping regime, $t_1/t_2 = 1$, and remains robust over a wide range of disorder strengths.

This pronounced spin transport stands in stark contrast to the behavior of the charge current in the same parameter regime (see Fig. 5(c)), where $I_{\mathrm{charge}}$ is strongly suppressed. The anti-correlation between spin and charge transport signals the emergence of a regime supporting nearly pure spin current where spin flow is decoupled from net charge motion. *Such a regime is of profound technological interest, offering a route to realizing low-dissipation spintronic functionalities in quasiperiodic quantum systems.* The underlying mechanism likely involves spin-dependent interference and the selective suppression of charge-carrying modes via destructive interference, while spin-polarized channels remain active and coherent. This highlights the potential for engineering charge-neutral spin currents via microscopic control over hopping parameters and spin populations.

To further elucidate the role of spin imbalance configuration, Fig. 6(b) presents the corresponding phase diagram for a second scenario (even down spin), where the distribution of spin-down electrons is altered. In this case, the spin current is markedly reduced across most of the explored parameter space, especially for moderate to strong disorder and throughout the $t_1/t_2$ spectrum. This contrast reveals that the nature of spin transport is acutely sensitive to the specifics of the spin imbalance namely, which spin species dominates. The diminished spin flow likely arises from reduced overlap in the spin-resolved density of states and weakened coherence among spin-polarized transmission paths.

Collectively, these findings establish that quasiperiodic systems with tailored spin imbalance and hopping asymmetry can host regimes of high-fidelity spin transport, even under substantial disorder. The ability to generate spin currents in the absence of net charge flow represents a critical advance toward spin-based logic and memory elements. Furthermore, the parameter-dependent tunability of spin current, via $t_1/t_2$, $\lambda_2$, and spin imbalance profile offers a flexible framework for designing reconfigurable, disorder-resilient spin transport devices. Such architectures could serve as the foundation for spin batteries, spin filters, and other nanoscale spintronic applications where energy efficiency and robustness against structural aperiodicity are essential.

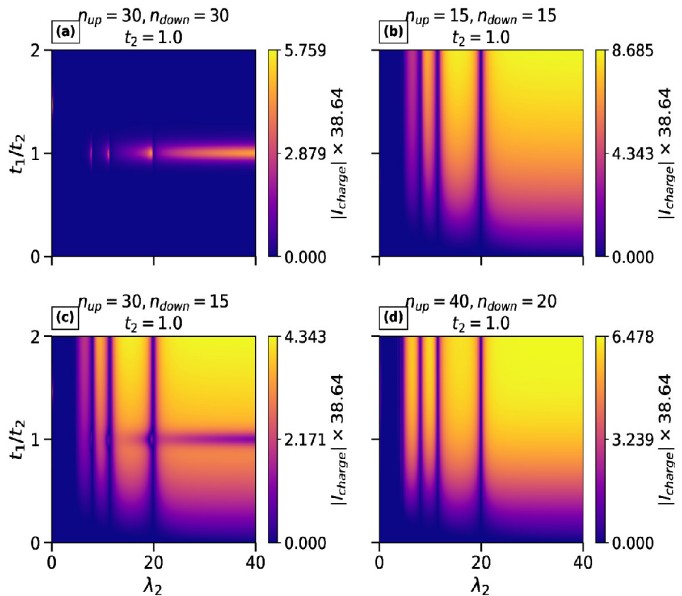

FIG. 7: (Color online). Charge current as a function of quasiperiodic strength $\lambda_2$ and hopping ratio $t_1/t_2$ at fixed phase $\phi_2 = \pi$ and Aharonov-Bohm flux $\theta = 0.3\,\theta_0$. (a) Half filling; (b) quarter filling; (c–d) spin-imbalanced configurations.

### E. Staggered Phase ($\phi_2 = \pi$): Charge and Spin Current Response to $t_1/t_2$ and $\lambda_2$

We now explore a strategically modified version of the original tight-binding framework, where the Aubry-André-Harper (AAH) phase exhibits an explicit site-dependent modulation within each unit cell. Concretely, we impose a phase configuration of $\phi_1 = 0$ and $\phi_2 = \pi$, thereby introducing a local staggered structure in the on-site potential across the two inequivalent sites. This internal phase asymmetry breaks the phase uniformity at the unit-cell level and significantly enriches the quasiperiodic landscape encountered by itinerant electrons. Such a configuration is not merely a minor adjustment; it fundamentally alters the interference conditions governing quantum transport. By shifting the relative phase between sublattices, we effectively engineer a novel class of quasiperiodic modulation that modifies both the energy spectrum and wavefunction profiles. This can lead to highly nontrivial localization phenomena and transport characteristics. The influence of this site-specific phase offset is comprehensively analyzed in Figs. 7 and 8, where we present the behavior of both charge and spin currents under various spin population scenarios including balanced and imbalanced configurations.

### 1. Charge transport characteristics under staggered AAH phase modulation

We begin by analyzing the behavior of the charge current, $I_{\mathrm{charge}}$, in the balanced spin configuration under the influence of staggered AAH phase modulation. Fig-

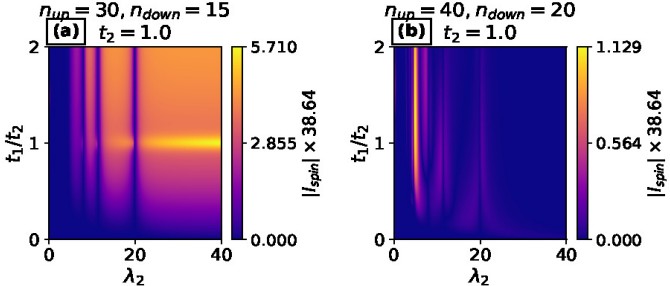

FIG. 8: (Color online). Spin current plotted as a function of the quasiperiodic modulation strength $\lambda_2$ and the hopping ratio $t_1/t_2$, with the Aubry phase fixed at $\phi_2 = \pi$ and $\theta = 0.3\,\theta_0$. Panels (a) and (b) correspond to cases with imbalanced spin populations.

ure 7(a) illustrates the half-filled scenario, where the number of up- and down-spin electrons are equal. Notably, the charge current reaches its peak around the homogeneous hopping regime, $t_1/t_2 \approx 1$, and remains appreciable over a wide range of disorder strengths $\lambda_2$. This enhancement signifies a robust coherence in transport when the hopping amplitudes are symmetric, even in the presence of substantial quasiperiodic potential.

However, as the system deviates from this symmetric point i.e., in the strongly dimerized regime $t_1 \neq t_2$ the current is markedly suppressed. This degradation of transport under increased hopping asymmetry mirrors the qualitative trends observed in the non staggered phase model (see Fig. 5(a)), though the absolute magnitude of the current is somewhat reduced due to the staggered phase profile. These observations underscore the delicate interplay between structural uniformity and phase disorder in regulating coherent charge dynamics.

At quarter-filling [Fig. 7(b)], the charge transport profile exhibits a distinctly different behavior. For disorder strengths $\lambda_2 \lesssim 10$, the current remains relatively low across the entire $t_1/t_2$ range than $\phi_2 = 0$ (non-staggered) case, suggesting the dominance of localization effects. Interestingly, beyond this threshold, a good amount of the charge current is observed, revealing signs of transport re-entrance. While this enhancement is modest compared to the corresponding case in the phase-homogeneous model (see Fig. 5(b)), the emergence of such re-entrant behavior highlights the nontrivial role of staggered phase potentials in shaping the electronic landscape which is also present for non-staggered case.

In Fig. 7(c), we investigate the charge transport behavior under spin-imbalanced conditions within the framework of the modified model featuring staggered phase modulation. While the qualitative features of the phase diagram characterized by regions of high and low current in the $(t_1/t_2, \lambda_2)$ plane closely mirror those observed in the earlier model (see Fig. 5(c)), the absolute magnitude of the charge current is notably diminished. This attenuation underscores the detrimental impact of spin population imbalance and staggered phase on coherent charge transport. More interestingly, the re-entrant transport characteristics, marked by multiple transitions (transi-

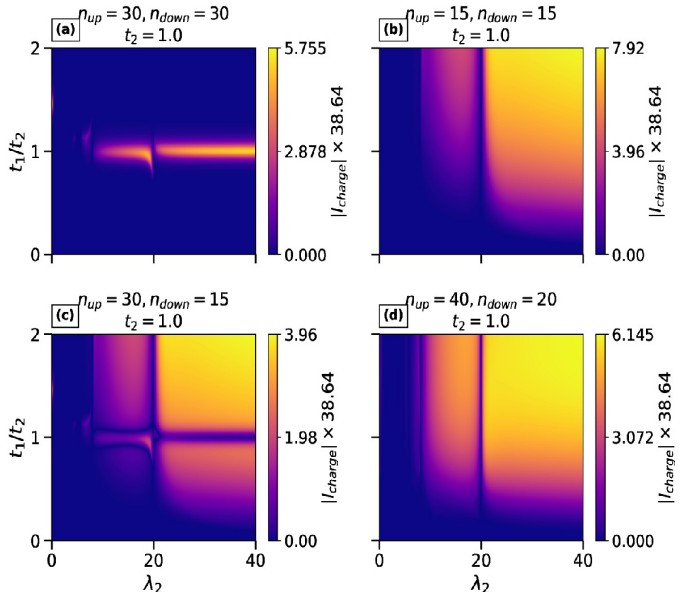

FIG. 9: (Color online). Charge current mapped over $\lambda_2$ and hopping ratio $t_1/t_2$ for $\phi_2 = \pi/2$ and fixed Aharonov-Bohm flux $\theta = 0.3\,\theta_0$. Panel (a) shows results at half-filling, (b) at quarter-filling, while (c) and (d) highlight spin-imbalanced cases.

tioning from yellowish to bluish and vice versa) between conductive and non-conductive regimes with varying $\lambda_2$, appear more prominently than in the previous configuration. This enhanced re-entrance reflects the intensified competition between hopping asymmetry, staggered phase $(\phi_2)$ modulation, and spin imbalance, giving rise to a richer transport landscape.

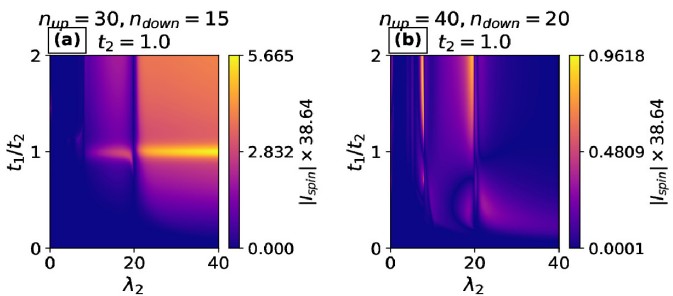

FIG. 10: (Color online). Variation of spin current with quasiperiodic modulation strength $\lambda_2$ and hopping ratio $t_1/t_2$ at fixed Aubry phase $\phi_2 = \pi/2$ and $\theta = 0.3\,\theta_0$. The plots highlight the impact of spin imbalance and phase engineering on spin transport under intermediate phase conditions.

A particularly striking feature emerging in this spin-imbalanced configuration is the appearance of narrow, vertically aligned low-current regions manifested as slender blue strips in the phase diagram that span the full range of the hopping ratio $t_1/t_2$. Compared to the analogous case in the earlier model (see Fig. 5(c)), these suppressed-current zones are considerably narrower, indicating that the parameter regime supporting finite charge transport is broader in the present setup. This effectively

translates to a wider window of disorder strengths over which conduction remains appreciable, suggesting enhanced robustness of transport against quasiperiodic perturbations under spin-imbalanced conditions. Moreover, when the spin-down electron population corresponds to an even configuration (see Fig. 7(d)), the system exhibits a distinctly pronounced re-entrant behavior, where the charge current undergoes multiple non-monotonic transitions with increasing $\lambda_2$. This observation points to a rich and intricate interplay between spin population imbalance, internal phase asymmetry, and quasiperiodic modulation yielding complex transport dynamics that persist even in regimes typically dominated by disorder-induced localization. Such tunable re-entrant features present compelling opportunities for engineering electronic states with tailored conduction properties in designer quantum lattices.

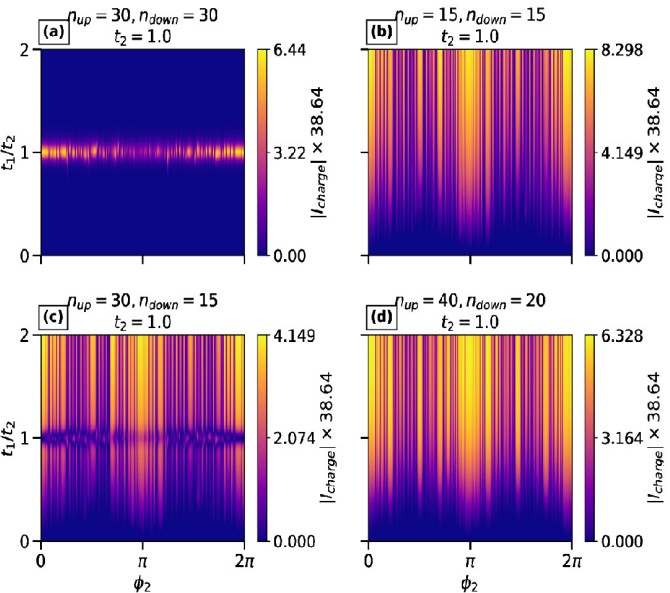

FIG. 11: (Color online). Charge current as a function of the hopping ratio $t_1/t_2$ and AAH phase $\phi_2$, with the Aharonov-Bohm flux fixed at $0.3\,\theta_0$. Panel (a) corresponds to the half-filled case, while panel (b) presents results for the quarter-filled configuration. Panels (c) and (d) explore scenarios with spin-imbalanced populations. Results are shown for $\lambda_2 = 22$.

### 2. Spin current characteristics under staggered phase

The behavior of spin current in the presence of spin imbalance reveals strikingly selective transport regimes, intimately tied to the parity of the spin population. As shown in Fig. 8(a), for configurations with an odd number of down-spin electrons, the spin current profile closely mirrors that of the charge current over much of the parameter space, with a remarkable deviation emerging near the symmetric hopping regime ($t_1 \approx t_2$) under strong disorder ($\lambda_2 \gtrsim 20$). In this regime, where charge current is nearly extinguished due to pronounced localization, the spin current remains robust and even exhibits

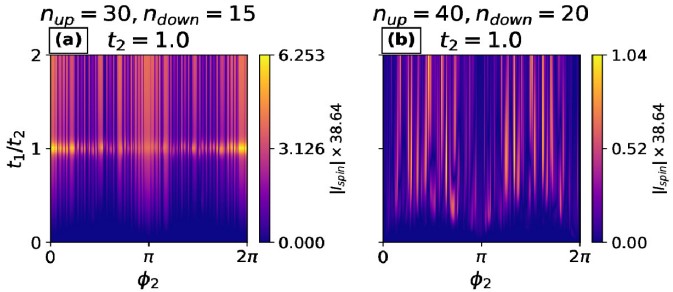

FIG. 12: (Color online). Spin current mapped over the hopping ratio $t_1/t_2$ and the AAH phase $\phi_2$, with the Aharonov–Bohm flux fixed at $0.3\,\theta_0$. Panels (a) and (b) correspond to spin-imbalanced populations, evaluated at $\lambda_2 = 22$.

notable enhancement. This signals the emergence of a charge-neutral spin transport channel a hallmark of dissipationless spin flow suggesting that spin coherence can persist in the absence of accompanying charge dynamics. In sharp contrast, Fig. 8(b) shows that for an even number of down-spin electrons, the spin current remains uniformly suppressed across the maximum landscape of $t_1/t_2$ and $\lambda_2$, even in regions where charge conduction survives. This asymmetry strongly indicates that spin transport is sensitive not just to hopping symmetry and disorder, but also to the parity of spin imbalance, which may influence interference conditions and the availability of spin-resolved pathways. These results establish practical design principles for achieving robust spin transport: in particular, odd-spin configurations tuned close to hopping symmetry and specific disorder window emerge as an optimal regime for realizing spin currents that are largely insensitive to charge fluctuations.

### F. Transport Characteristics Under Intermediate AAH Modulation ($\phi_2 = \pi/2$)

We now turn our attention to an intermediate phase configuration, where the Aubry-André-Harper (AAH) phases on the two sublattice sites within each unit cell are set to $\phi_1 = 0$ and $\phi_2 = \pi/2$, respectively. This site-specific phase difference introduces a tunable asymmetry in the quasiperiodic modulation, reshaping the underlying interference patterns and thereby modifying the charge and spin transport properties. We perform a comprehensive analysis of both current components under balanced and imbalanced spin configurations, exploring their dependence on hopping dimerization ($t_1/t_2$) and the quasiperiodic disorder strength ($\lambda_2$).

*Charge current:* The balanced spin case reveals a notable suppression in both the magnitude and robustness of transport, in contrast to the behavior observed in the non-staggered phase. Although the overall current profiles for half-filled and quarter-filled configurations retain qualitative resemblance to earlier results, the parameter space supporting appreciable charge flow narrows significantly (Fig. 9). In particular, the quarter-filled case shows suppressed conduction across much of the disorder

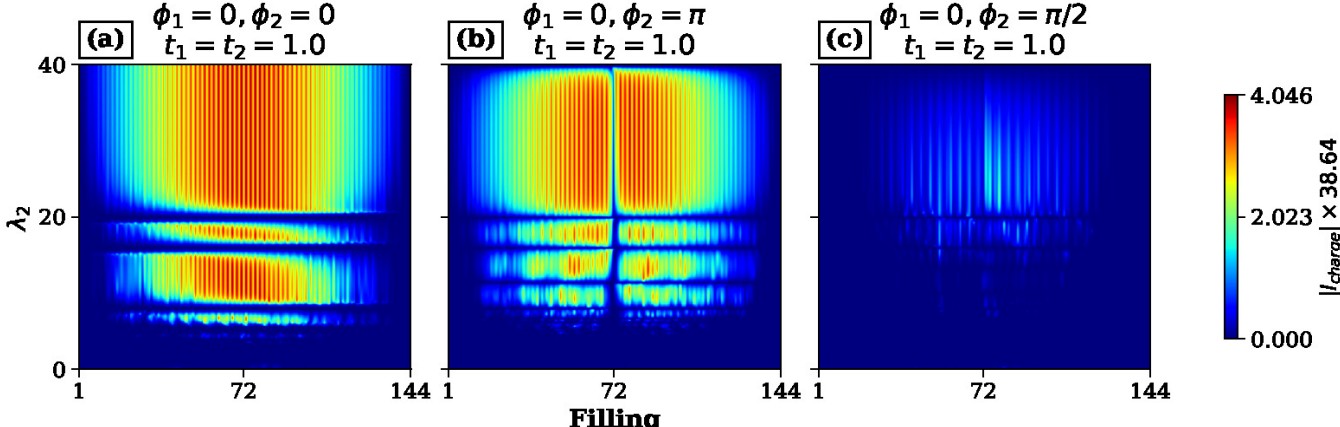

FIG. 13: (Color online). Charge current as a function of filling factor and quasiperiodic strength $\lambda_2$ for uniform hopping ($t_1 = t_2 = 1$) at $\theta = 0.3\,\theta_0$. Panel (a): symmetric phase configuration ($\phi_1 = \phi_2 = 0$); (b): staggered phase ($\phi_1 = 0$, $\phi_2 = \pi$); (c): intermediate phase ($\phi_1 = 0$, $\phi_2 = \pi/2$). Here we take 144 sites or 72 unit cell.

spectrum, indicating increased localization. Meanwhile, the half-filled scenario continues to exhibit moderate resilience like the earlier scenarios, suggesting that filling plays a nontrivial role in mitigating localization. When spin imbalance is introduced, prominent current peaks emerge near the symmetric hopping point ($t_1/t_2 \approx 1$) within the intermediate disorder window ($\lambda_2 \sim 10$–$20$) for odd down-spin electron numbers, along with a secondary enhancement in the regime $t_1 > t_2$ and $\lambda_2 \sim 20$–$40$. In contrast, the even-spin case aligns more closely with previous trends, dominated by localization-induced current suppression outside optimal transport corridors.

*Spin current:* The underlying dynamics present compelling opportunities for realizing spintronic functionalities. For odd spin-imbalance configurations, we observe a distinct enhancement of spin transport in the region $\lambda_2 \sim 20$–$40$, even as charge current dwindles to negligible levels (Fig. 10). This selective activation of spin channels underpins a potential regime of dissipationless spin flow, facilitated by spin-dependent interference and minimal backscattering that is inaccessible via conventional charge transport mechanisms. Notably, such behavior is largely absent for even down-spin populations, underscoring the crucial influence of spin parity on the emergence of pure spin currents. These results not only validate the robustness of spin transport under quasi-periodic modulation but also position the $\phi_2 = \pi/2$ model as a promising candidate for reconfigurable and low-power spintronic architectures.

### G. Phase-engineered control of charge transport: Variation with AAH phase

In Fig. 11, we present a comprehensive phase diagram of the charge current as a function of the hopping ratio $t_1/t_2$ and the secondary AAH phase $\phi_2$, with the primary phase fixed at $\phi_1 = 0$. This landscape captures the intricate interplay between quasiperiodicity, dimerization, and spin configuration, revealing a variety of transport regimes. For half-filled systems, maximal current emerges near the homogeneous hopping limit ($t_1/t_2 \approx 1$), where the suppression of dimerization-induced gaps allows the wavefunctions to remain largely extended. However, a dramatically different scenario unfolds in the quarter-filled configuration and in spin-imbalanced systems with even down-spin populations: here, robust current persists over a broader range of both $t_1/t_2$ and $\phi_2$, indicating a large swath of the phase space supports extended states. This reveals that partial filling and appropriate spin configurations can cooperate with phase modulation to create a stable transport channel across wide disorder and dimerization conditions. For the case of odd down-spin occupation, the current exhibits a noticeable reduction near $t_1/t_2 \approx 1$ across a broad range of $\phi_2$, except at $\phi_2 = \pi$. In general, the current attains relatively higher values in the vicinity of $\phi_2 = 0$, $\pi$, and $2\pi$, compared to other values of $\phi_2$.

In Fig. 12, we present the phase diagrams of the spin current as a function of the AAH phase $\phi_2$ and the hopping ratio $t_1/t_2$. A striking feature emerges for the case of odd down-spin filling: a prominent yellow strip appears around $t_1/t_2 \approx 1$ [see Fig. 12(a)], spanning almost the entire $\phi_2$ window ($0 \le \phi_2 \le 2\pi$). This bright region signifies a regime of enhanced spin transport, where spin current reaches its maximum response. Interestingly, in the same filling sector the charge current exhibits a contrasting behavior, marked by a blue strip [Fig. 11(c)], indicating a substantial suppression. Consequently, the spin current overwhelmingly dominates the charge current across a wide range of $\phi_2$ values near $t_1/t_2 \sim 1$. In sharp contrast, for even down-spin filling the scenario is reversed: the spin current is weaker than the charge current [see Fig. 12(b)], underscoring the crucial role of spin imbalance in driving robust spin transport in this quasiperiodic ring geometry.

It is worth emphasizing that our essential findings primarily focus on $\phi_2 = 0$, $\pi$, and $\pi/2$, since at these points the cosine modulation respectively remains unchanged, changes sign, or transforms into a sine function. Nev-

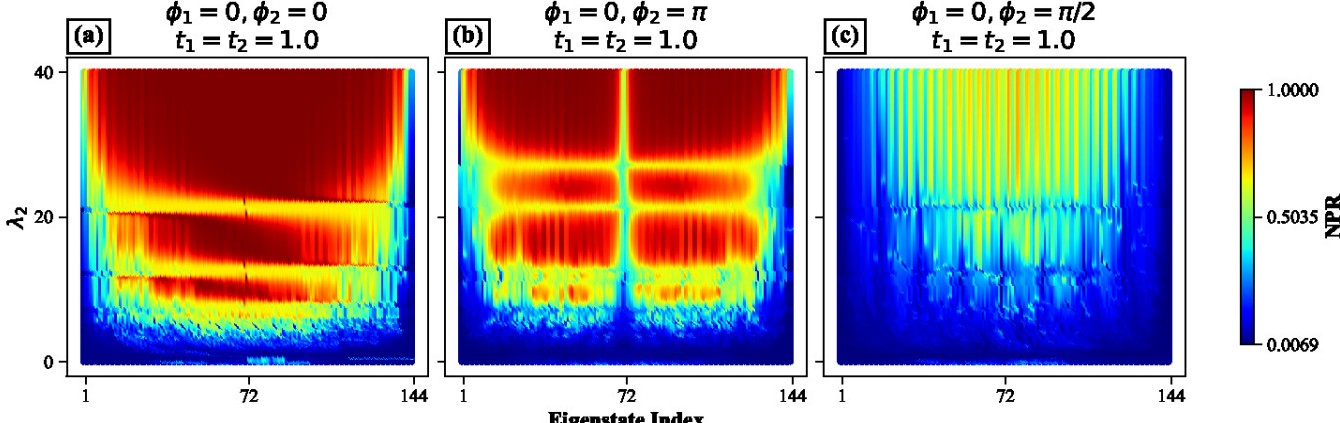

FIG. 14: (Color online). Nonmalized participation ratio (NPR) plotted as a function of eigenstate index and quasiperiodic modulation strength $\lambda_2$ for the uniform hopping case $t_1 = t_2 = 1$ with $\theta = 0.3\,\theta_0$. Panel (a) illustrates the non-staggered phase configuration with $\phi_1 = \phi_2 = 0$, while panels (b) and (c) explore staggered and intermediate cases with $\phi_1 = 0$, $\phi_2 = \pi$ and $\phi_1 = 0$, $\phi_2 = \pi/2$, respectively. Given the similarity in NPR between up and down spin sectors, an average over both is used to represent the localization characteristics. System size is same as earlier plot.

ertheless, by presenting the continuous variation of $\phi_2$ over the interval $[0, 2\pi]$, we highlight the global influence of $\phi_2$ on the current profile. Importantly, if one wishes to explore alternative choices of $\phi_2$, there always exists a suitable set of values yielding appreciable current. Altogether, these findings underscore the utility of the AAH phase $\phi_2$ as a powerful control knob for engineering transport properties. By finely tuning the dimerization and phase parameters, one can dynamically modulate the conduction behavior across various filling and spin configurations, paving the way for design strategies aimed at spin- and phase-controlled electronic functionalities in correlated quasiperiodic systems.

## H. Filling-tuned re-entrant transport transitions

In Fig. 13, we explore the dependence of the charge current on the secondary quasiperiodic potential strength $\lambda_2$ and the electronic filling factor, across three representative models with distinct AAH phase configurations $(\phi_1, \phi_2)$. The analysis is performed under equal hopping amplitudes $(t_1 = t_2)$, which maximizes the propensity for delocalized transport. Remarkably, the current exhibits a highly non-monotonic behavior with $\lambda_2$ and filling, featuring a series of re-entrant transitions where the system alternates between current-carrying and suppressed regimes. These recurring finite-current windows indicate the emergence of extended-state islands embedded in an otherwise localized landscape, controlled not only by the potential strength but also by the electronic filling. At low fillings, the current remains small, increases with higher filling, and eventually drops to zero as the system approaches full filling. Notably, for the staggered phase configuration, a suppression of current is observed near half filling. Overall, intermediate-phase setups yield reduced current, while non-staggered configurations maximize transport, highlighting the electronic

filling as a tunable parameter for accessing unconventional transport regimes in quasiperiodic systems.

## I. Variation of NPR resolved eigenstate index with $\lambda_2$

Figure 14 presents a comprehensive map of the normalized participation ratio (NPR) across the full single-particle eigenstate spectrum as a function of the quasiperiodic potential strength $\lambda_2$, for three representative phase configurations $(\phi_1, \phi_2)$ under symmetric hopping $(t_1 = t_2)$ and system size $L = 144$. As a quantitative measure of delocalization, the NPR provides crucial microscopic insight into the nature of eigenstates, with high values signaling extended states conducive to transport, and low values indicating localized, non-conducting modes. The resulting color density plots reveal rich spectral patterns characterized by multiple delocalization-localization transitions as $\lambda_2$ is tuned, exhibiting striking correspondence with the re-entrant features observed in the filling-dependent current profiles (Fig. 13). Particularly noteworthy is the emergence of successive re-entrant regimes along the $\lambda_2$ axis, where the system alternates between extended (dark red) and relatively localized (yellowish) eigenstates, highlighting the non-monotonic character of localization in quasiperiodic systems. This direct correlation between NPR fluctuations and macroscopic transport behavior offers compelling microscopic validation of the dynamical transport landscape, highlighting the critical role of quasiperiodic phase modulation in engineering tunable, nontrivial conducting phases in low-dimensional correlated systems.

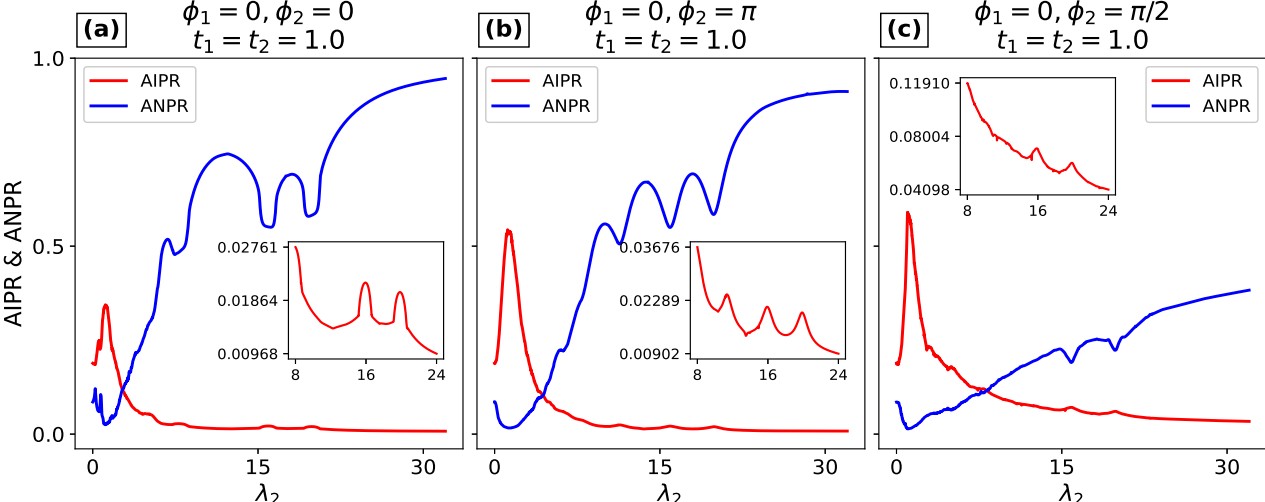

FIG. 15: (Color online). Evolution of average inverse participation ratio (AIPR) and average normalized participation ratio (ANPR) as a function of the quasiperiodic modulation strength $\lambda_2$ in the symmetric hopping configuration $t_1 = t_2 = 1$ at $\theta = 0.3\,\theta_0$. Panel (a) corresponds to the fully phase-aligned setup with $\phi_1 = \phi_2 = 0$. Panels (b) and (c) explore the impact of introducing asymmetry through the relative Aubry phase differences, with $\phi_1 = 0$, $\phi_2 = \pi$ and $\phi_1 = 0$, $\phi_2 = \pi/2$, respectively. Since both spin channels yield similar average NPR and IPR values, we use their mean to simplify the analysis. The system size is taken to be $N = 144$ for the presented data.

## J. Variation of average IPR and NPR with $\lambda_2$

In Fig. 15, we systematically investigate the global localization properties of the system by analyzing two key statistical indicators, AIPR and ANPR, as functions of the secondary quasiperiodic potential strength $\lambda_2$. This analysis is carried out for three distinct models, each characterized by different phase configurations $(\phi_1, \phi_2)$, under the symmetric hopping condition $t_1 = t_2$, which earlier emerged as the regime exhibiting the most pronounced charge transport. The AIPR serves as a diagnostic of wavefunction localization, with larger values indicating enhanced spatial confinement, while the ANPR reflects the average extent of delocalization across the spectrum. As shown, both measures exhibit a highly non-monotonic dependence on $\lambda_2$, revealing a series of localization-delocalization transitions that mirror the complex transport landscape observed in previous sections. Notably, non-staggered phase set up consistently shows lower AIPR and higher ANPR values over a broader $\lambda_2$ range, reaffirming its superior transport characteristics, likely due to more favorable interference conditions. In contrast, the intermediate phase configuration is characterized by a higher AIPR and lower ANPR compared to the other phase settings, indicating a stronger suppression of delocalized states and, consequently, of transport. These statistical trends provide compelling evidence of a strong correlation between eigenstate morphology and macroscopic current behavior, and underscore the critical role of quasiperiodic phase engineering in modulating transport efficiency. Our findings thus establish AIPR and ANPR as robust, complementary metrics for characterizing localization physics in phase-tunable quasiperiodic lattices.

Both AIPR and the ANPR serve as global diagnostics for the spatial character of eigenstates across the full spectrum. While the AIPR measures the degree of localization where higher values signal stronger confinement the ANPR captures the extent of delocalization, with larger values indicating more extended wavefunctions. As the strength of the secondary quasiperiodic potential $\lambda_2$ is tuned, we observe a sequence of non-monotonic variations in both AIPR and ANPR, signaling multiple localization–delocalization (LD) and delocalization–localization (DL) transitions. These re-entrant spectral features, which emerge at several critical values of $\lambda_2$, align closely with sharp transport transitions previously identified near $\lambda_2 \approx 20$, reinforcing the intrinsic link between microscopic eigenstate morphology and macroscopic current behavior. While these statistical quantities are averaged over the entire eigenspectrum unlike the current, which is computed at fixed filling fractions such as half-filling, quarter-filling, and spin-imbalanced cases their qualitative agreement with transport signatures underscores their value as predictive indicators.

To understand the scenario more precisely let us consider the case $\phi_1 = \phi_2 = 0$, for which the site potential takes the form $\epsilon_i = \lambda_1 \cos(2\pi i b)/(1 + \lambda_2 \cos(2\pi i b) + \eta)$, applied unit-cell-wise such that two sites within a unit cell share the same index. We observe a non-monotonic behavior of ANPR as a function of $\lambda_2$. At small $\lambda_2$, the potential is mainly governed by the numerator $\lambda_1 \cos(2\pi i b)$, which provides only moderate modulation, resulting in localized states with ANPR values significantly below unity and approaching zero. As $\lambda_2$ increases to intermediate values, the denominator introduces nonlinear variations in the effective site energies, producing local energy landscapes where some states are more localized and

| Hopping configuration | Non staggered phase $\phi_1 = 0,\ \phi_2 = 0$ | Staggered phase textbf $\phi_1 = 0, \phi_2 = \pi$ | Intermediate phase $\phi_1 = 0\ ,\ \phi_2 = \pi/2$ |
|---|---|---|---|
| $t_1 = t_2$ | • Densely packed energy flux bands with notably high NPR values, signaling strong conducting behavior. • Non-monotonic current profile with $\lambda_2$ reveals multiple extrema, reentrant features, and eventual stabilization for both half- and quarter-filling. • For odd down-spin sectors, spin current maximizes near $t_1 = t_2$, while charge current minimizes, marking low dissipation spin transport pathways. | • Small gap emerges near zero energy, with lower NPR than non-staggered case. • Current saturates at higher $\lambda_2$, signaling delayed stabilization compared to non-staggered phase. • Charge and spin transport moderately suppressed; conducting zone in $t_1/t_2$–$\lambda_2$ space shrinks, although showing sharper re-entrant transport signatures. | • Flat bands with near-zero NPR emerge at spectral edges, enhancing localization. • Compared to the staggered case, current at half-filling is enhanced with increasing $\lambda_2$, while it stays suppressed at quarter-filling, a reversed trend indicating phase-selective transport. • The spin current dominates over the charge current, rendering the system dissipationless for $\lambda_2 \geq 20$. |
| $t_1 > t_2$ | • Energy flux landscape shows clear band gap, reduced current, and isolated peaks at half-filling; quarter-filled case almost mimics the $t_1 = t_2$ configuration. • Current at half-filling shows no saturation trend as $\lambda_2$ increases steadily. • Spin imbalance causes suppression of spin current than $t_1 = t_2$ case, limiting dissipationless spin flow possibilities. | • Band gap increases moderately with lower NPR values than the non-staggered case. • Charge current stays minimal at half-filling, becomes notable in quarter-filled regime; spin current also reduces. | • Band gap reduces compared to staggered case, with enhanced current at half-filling than staggered case. • Spin current shows modest suppression compared to staggered phase configuration. |
| $t_1 < t_2$ | • Increased band gap suppresses current at half filling. • While both this and staggered setups show a current dip with $\lambda_2$, the intermediate phase strikingly features a pronounced peak around the same point [see Fig. 4(c)]. • Charge and spin currents are consistently lower than the $t_1 > t_2$ configuration. | • At half filling, current shows nonmonotonic trend with $\lambda_2$: it rises, drops, and revives at higher modulation strengths. • This hopping configuration is less efficient than the non-staggered setup for both charge and spin transport response. | • At half-filling, a pronounced peak appears in the current versus $\lambda_2$, with its magnitude exceeding that of the other phase configurations [see Fig. 4(c)]. • For the quarter-filling case, the current remains lower than in the other two configurations. • This setup exhibits the weakest spin transport among all compared phase configurations. |

TABLE I: A systematic comparison of charge and spin transport responses across three distinct quasiperiodic phase configurations $(\phi_1, \phi_2)$, demonstrating how engineered phase asymmetries selectively modulate transport channels for the three different hopping correlations.

others more extended, resulting in oscillations in ANPR values. In this regime, the unit-cell-wise potential allows intra-cell resonance, which competes with the site energy variations and generates the observed fluctuations. For large $\lambda_2$, the denominator dominates, effectively flattening the potential across all sites, which enhances intra-cell resonance and leads to fully extended states, causing ANPR to saturate at unity. Thus, the re-entrant behavior of ANPR and its eventual saturation can be physically understood as the result of the *interplay between numerator and denominator in the site potential, combined with the intra-cell structure that favors delocalization at large $\lambda_2$*. For AIPR, the intermediate oscillations are less pronounced than in the ANPR case (see insets of Fig. 15), yet the trend remains evident in the ANPR analysis, underscoring its higher sensitivity to the localization–delocalization crossover in this model and its consistency with the transport behavior.

### K.   Justification of our model

If we consider a ring with random or Aubry–André–Harper (AAH) type site-energy modulation, the persistent current decreases monotonically with increasing disorder strength and eventually vanishes in the strong disorder limit. Inclusion of next-nearest-neighbor hopping (NNNH) can lead to a slight enhancement of the current compared to the nearest-neighbor hopping (NNH) case, but the overall trend with increasing disorder strength remains decreasing. On the other hand, when the model is restricted to NNH only and a unit-cell modulation is introduced such that two sites within a unit cell share the same site index, with either staggered or non-staggered potential variation, the current exhibits a nonmonotonic behavior: it initially increases with disorder strength and then decreases again for certain correlations between intra- and inter-unit-cell hoppings. This single-peak feature of the current appears at half-filling, naturally raising the question of whether multiple such peaks can be realized. Furthermore, in earlier studies the stability of the current could not be achieved, as the current always decayed to zero with increasing disorder strength without reaching a finite constant value. In contrast, in our work we employ a modified model that simultaneously yields both multiple current peaks and current stabilization, where the current saturates to a finite nonzero value in the strong disorder regime. We also demonstrate that this scenario, while observed at half-filling, remains valid for the quarter-filling case as well..

In this model, we consider two types of Aubry–André–Harper (AAH) phases, $\phi_1$ and $\phi_2$, and focus on three representative choices for $\phi_2$ while keeping $\phi_1 = 0$, namely $\phi_2 = 0, \pi, \pi/2$, which provide distinct physical scenarios. For $\phi_2 = 0$, the denominator of the site potential for both sites in a unit cell becomes $(1+\lambda_2 \cos(2\pi ib))$, leading to a non-staggered configuration in which the two sites within a unit cell have identical site energies. When $\phi_2 = \pi$, one site retains $(1 + \lambda_2 \cos(2\pi ib))$, while the other becomes $(1-\lambda_2 \cos(2\pi ib))$, since $\cos(A_i+\pi) = -\cos A_i$ with

$A_i = 2\pi ib$, resulting in a staggered configuration where the effective site energies differ within the same unit cell despite identical site indices. For $\phi_2 = \pi/2$, one site has $(1+\lambda_2 \cos(2\pi ib))$ and the other $(1-\lambda_2 \sin(2\pi ib))$, providing an intermediate configuration in which the site energies within a unit cell are neither identical nor strictly opposite, but follow distinct trigonometric forms. The consideration of these three phase choices allows a systematic investigation of the interplay between non-staggered, staggered, and mixed (cosine-sine) potential landscapes, enabling the study of how intra- and inter-unit-cell correlations, as well as relative phase differences, influence localization, delocalization, and transport properties.

### L.   Experimental realization

Experimental studies on persistent currents in mesoscopic rings and topological phase transitions in SSH-like systems[75] have laid the foundation for realizing quasiperiodic lattice models. Our model, which incorporates a generalized AAH potential, can be implemented in ultracold atom systems using bichromatic optical lattices, where quasiperiodicity and the AAH phase $\phi$ can be tuned by controlling the relative phase or displacement between two incommensurate laser fields. Ring-like geometries can be engineered using optical tweezers or spatial light modulators, while modulated hopping amplitudes, essential for SSH-type structures, can be achieved by alternating site spacings or using atoms with different properties. Persistent currents [76] may then be probed using synthetic gauge fields or interferometric methods. Detection of the current can be achieved via measurement of the induced magnetic dipole moment using SQUID instrumentation. This setup offers a feasible platform for realizing stable quantum transport and exploring applications in atomtronics and spin-based quantum devices.

### IV.   CONCLUSIONS

In this work, we have presented a comprehensive numerical investigation of flux-induced transport in a quasiperiodically modulated SSH-type tight-binding ring threaded by an Aharonov-Bohm flux. By systematically exploring the interplay among hopping dimerization, secondary quasiperiodic potential strength, phase set up and electronic filling, we unveil a rich array of charge and spin transport phenomena. Our key findings are summarized below:

- **Re-entrant transport and localization transitions:** A hallmark of our results is the discovery of multiple re-entrant transitions between localized and extended states as the quasiperiodic potential strength $\lambda_2$ is tuned. These transitions are sharply reflected in both the persistent current response and the eigenstate character, as captured by inverse and normalized participation ratios (IPR and NPR).

- **Phase-engineered transport control:** The transport characteristics are strongly influenced by the phase configuration of the secondary incommensurate potential. A uniform, non-staggered phase yields the largest charge currents, particularly in the regime $t_1 \gtrsim t_2$, while the intermediate configuration with a relative phase of $\pi/2$ leads to a pronounced suppression of transport at both half and quarter fillings. The spin current exhibits an analogous behavior, underscoring the essential role of on-site phase modulation in shaping interference-induced localization landscapes.

- **Filling-tuned sensitivity and spin-resolved dynamics:** Quarter filling emerges as a particularly favorable regime, where re-entrant transport features become more prominent, providing a richer platform for exploring correlated transport phenomena. In addition, spin-imbalanced configurations, most notably those with an odd number of down spins, exhibit pronounced spin-current responses, including regimes near $t_1 = t_2$ where spin transport persists even in the absence of net charge flow. This highlights the potential of such configurations as viable candidates for spin-filtering applications.

- **Disorder-assisted delocalization:** Interestingly, the system exhibits both suppression and enhancement of transport at intermediate disorder strengths, thereby challenging the conventional expectation of strictly monotonic localization behavior. This suggests a nontrivial disorder-induced delocalization mechanism unique to quasiperiodic systems under magnetic flux.

- **Stable Current Generation:** The current exhibits near saturation beyond a certain threshold of $\lambda_2$, particularly in the half-filled regime for $t_1 = t_2$, and consistently across all hopping configurations ($t_1 = t_2$, $t_1 > t_2$, and $t_1 < t_2$) in the quarter-filled case. This robust saturation behavior indicates a remarkable degree of stability in the transport response, largely insensitive to further increases in the quasiperiodic modulation strength. Such a feature is highly desirable for practical applications, as it enables the system to function as a reliable constant current source. This stability under varying system parameters also opens up potential avenues for designing quantum devices where consistent current output is crucial, such as in nanoscale current regulators, precision metrology, and quantum information processing platforms that demand low-noise, tunable current sources. Furthermore, the tunability of saturation thresholds via $\lambda_2$ and filling fractions offers an additional degree of control, enhancing the flexibility and functionality of the system for spintronic and electronic circuit applications.

- **Spectral diagnostics and transport correspondence:** Although AIPR and ANPR do not always offer a strict one-to-one mapping with transport signatures due to their global nature, the observed trends in these spectral measures qualitatively corroborate the transport behavior and serve as effective microscopic indicators of extended versus localized regimes.

- **Implications for spintronic architectures:** The emergence of finite spin currents in the absence of charge transport particularly around the balanced hopping regime ($t_1 \approx t_2$) under odd spin imbalance points to a promising route for realizing energy-efficient, low-dissipation spintronic devices based on quasiperiodic interference control.

- **Model flexibility and feasibility of implementation:** The sensitivity of the model to tuning parameters such as $t_1/t_2$, $\lambda_2$, electron filling, and spin configuration renders it a highly flexible framework for designing quantum coherent transport devices. The proposed setup can be feasibly engineered in cold-atom lattices, photonic waveguides, or synthetic quantum materials, offering a testbed for reconfigurable spin-charge separation and quasiperiodic control.

Taken together, our study not only deepens the theoretical understanding of flux-driven transport in quasiperiodic mesoscopic systems but also identifies actionable pathways toward experimental realization and functional device integration. These results lay the groundwork for future studies involving many-body effects, disorder-topology interplay, and finite-temperature spin caloritronics.

## DATA AVAILABILITY STATEMENT

The data supporting the conclusions of this work are available from the authors upon reasonable request.

## DECLARATION

**Conflict of Interest:** The authors declare that there are no conflicts of interest related to this work.

### Appendix A: NPR resolved state current

In Fig. 16, we show the NPR-resolved state current as a function of the corresponding state index for two different values of $\lambda_2$, namely $\lambda_2 = 15$ (top row) and $\lambda_2 = 20$ (bottom row), corresponding to high and low magnitudes of the state current, respectively. The three columns correspond to $\phi_2 = 0$, $\pi$, and $\pi/2$, respectively. For $\lambda_2 = 15$, the state current amplitudes are larger, consistent with higher NPR values and a column-wise comparison reveals that in the third row corresponding to $\phi_2 = \pi/2$ [Fig. 16(c)], the current is reduced compared to the other two cases. Moreover, the NPR is also smaller in this case,

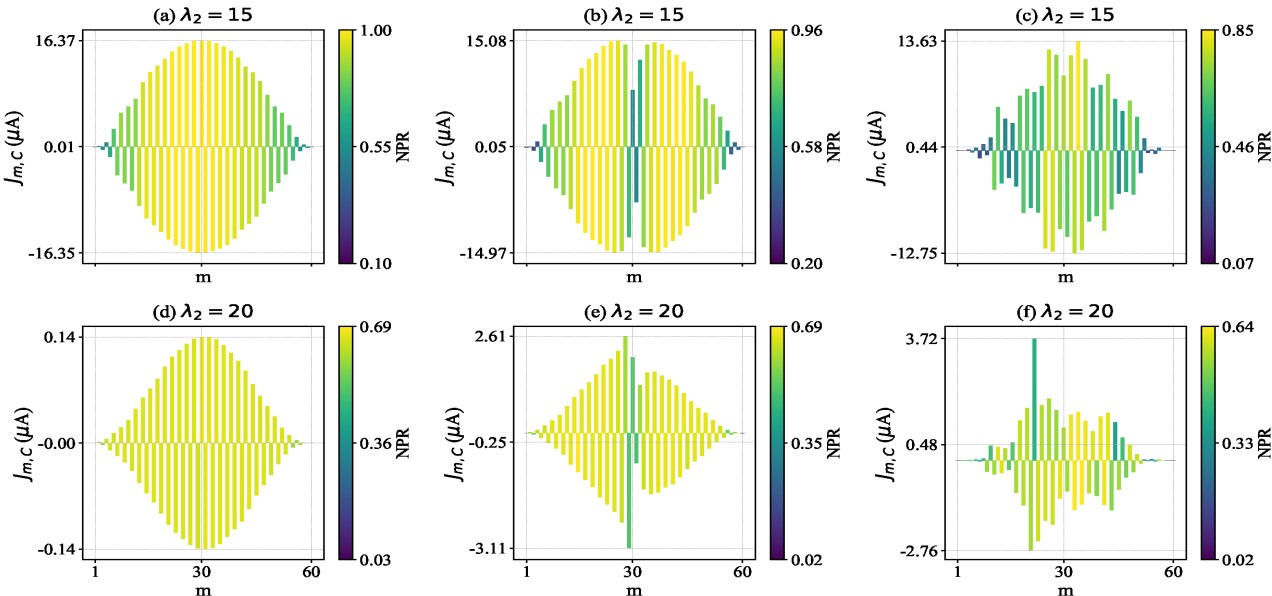

FIG. 16: (Color online). NPR-resolved state currents are plotted as a function of the eigenstate index for $\lambda_2 = 15$ (upper row) and $\lambda_2 = 20$ (lower row), corresponding to a 60-site (30-unit-cell) ring with $t_1 = t_2 = 1$, $\lambda_1 = 1.5$, and $\theta = 0.3\theta_0$. The charge current for the $m^{\text{th}}$ eigenstate is defined as $J_{m,C} = J_{m,\uparrow} + J_{m,\downarrow}$. Each column corresponds to three different values of $\phi_2$, namely 0, $\pi$, and $\pi/2$, respectively. Since the NPR is identical for equal number of spin-up and spin-down electrons, we take the spin-averaged values of NPR.

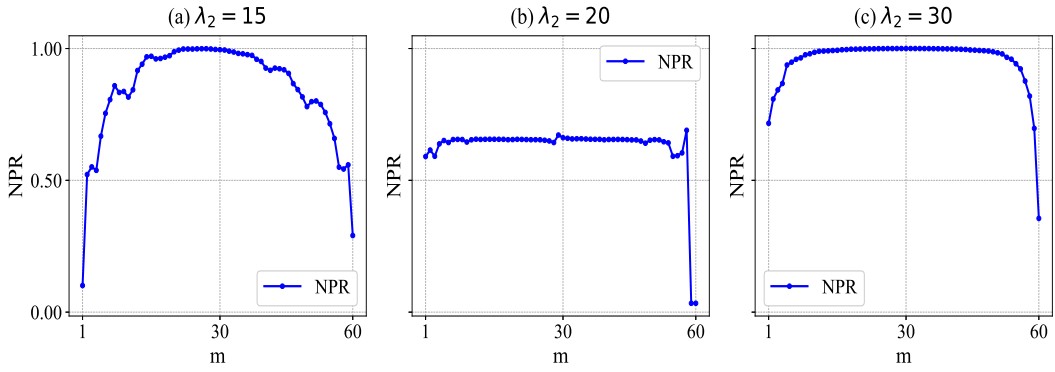

FIG. 17: (Color online). Individual NPR with corresponding state index $m$ for three representative $\lambda_2$ values at $\phi_2 = 0$, $\theta = 0.3\,\theta_0$, $t_1 = t_2$ and $\lambda_1 = 1.5$ in a 60-site ring.

with many states exhibiting lower NPR values. This further indicates that $\phi_2 = \pi/2$ corresponds to suppressed charge transport. For $\lambda_2 = 20$, both the current and the NPR are reduced much compared to the $\lambda_2 = 15$ case, irrespective of the chosen phase configuration. The edge states exhibit smaller NPR values, indicating that they are relatively more localized than the bulk states near the center of the spectrum. Although mutual cancellation occurs between states carrying positive and negative currents when summed up to half filling, the overall current remains significant for $\lambda_2 = 15$ due to the large magnitude of individual state currents, whereas for $\lambda_2 = 20$, the currents are sufficiently small that the net current after cancellation is negligible. This demonstrates a direct correlation between NPR and the magnitude of the state current: for $\lambda_2 = 15$, many states exhibit high NPR with peaks exceeding 0.8, while for $\lambda_2 = 20$, the NPR peaks

are generally lower, around 0.6, illustrating that higher NPR generally corresponds to larger contributions to the state current.

## Appendix B: NPR with state index

In Fig. 17, we show the NPR as a function of the state index for three representative values of $\lambda_2$, with $\phi_2 = 0$. The plot reveals that for $\lambda_2 = 15$, many states exhibit high NPR values approaching 1, indicating that these states are nearly fully extended. This trend becomes even more pronounced for $\lambda_2 = 30$, where an even larger fraction of states attain NPR close to 1, suggesting strong delocalization. Consequently, one expects higher charge transport in these regimes, consistent with the observed

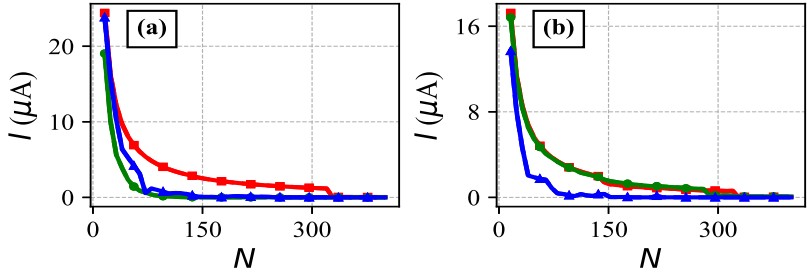

FIG. 18: (Color online). Variation of charge current with system size for $\lambda_2 = 15$, $\lambda_1 = 1.5$, $t_1 = t_2$, and $\theta = 0.3\,\theta_0$, shown for half filling (first column) and quarter filling (second column). The red, green, and blue curves correspond to $\phi_2 = 0$, $\pi$, and $\pi/2$, respectively.

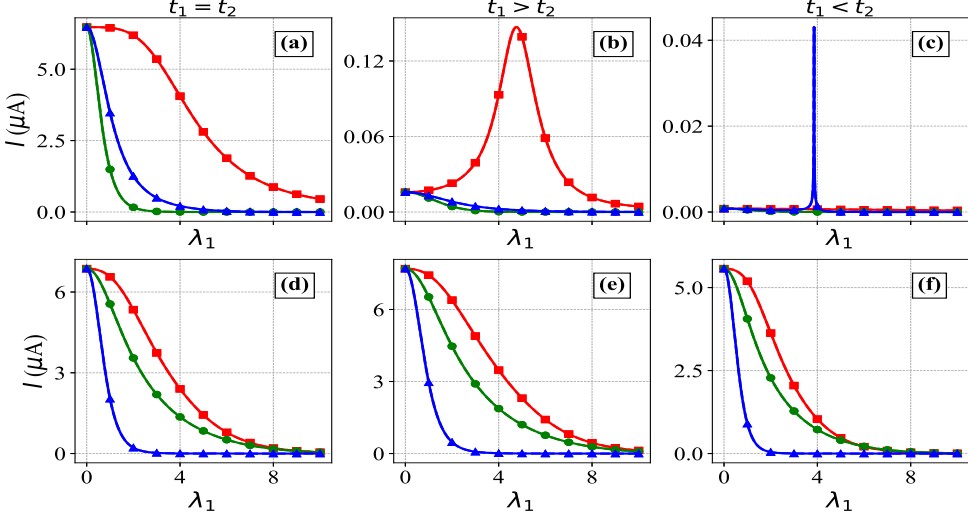

FIG. 19: (Color online). Variation of charge current with $\lambda_1$, with $\lambda_2$ fixed at 10 and $\theta = 0.3\theta_0$, shown for three hopping correlation cases as indicated in the plot. The red, green, and blue curves correspond to $\phi_2 = 0$, $\pi$, and $\pi/2$, respectively. The top row represents half filling, while the bottom row corresponds to quarter filling. The system size fixed at 60 sites.

current variations in the disordered system. In contrast, for $\lambda_2 = 20$, the majority of states display lower NPR values around 0.67, significantly below 1, implying that these states are less extended. This reduced spatial delocalization correlates with the smaller current observed in this case, further supporting the connection between NPR and charge transport in the system.

### Appendix C: Current with system size

In Fig. 18, we present the variation of charge current with system size for three representative values of $\phi_2$, as indicated in the figure caption. At half filling [Fig. 18(a)], the current is consistently higher for $\phi_2 = 0$ and $\pi/2$ compared to $\phi_2 = \pi$ across all system sizes. In contrast, at quarter filling, the current is larger for $\phi_2 = 0$ and $\pi$, while it is suppressed for $\phi_2 = \pi/2$. This indicates that the roles of $\phi_2 = \pi$ and $\pi/2$ effectively interchange when moving from half to quarter filling, highlighting that the phase configuration, together with the electronic filling, plays a crucial role in determining the magnitude of per-

sistent current. Furthermore, as persistent current is an inherently quantum mechanical phenomenon that relies on phase coherence, its magnitude decreases with increasing system size, consistent with the expected reduction due to finite coherence length in larger systems. These observations underscore that both the choice of quasiperiodic phase and the filling fraction are decisive factors for sustaining robust charge transport across different system sizes.

### Appendix D: Current dependence on $\lambda_1$ (justification for the choice of $\lambda_2$ as important parameter)

In Fig. 19, we present the variation of charge current as a function of $\lambda_1$ for three different hopping correlation cases, represented along the three columns. The red, green, and blue curves correspond to $\phi_2 = 0$, $\pi$, and $\pi/2$, respectively, and results are shown for both half- and quarter-filling conditions. For half filling, when $t_1 = t_2$, the current exhibits a monotonically decreasing

trend with increasing $\lambda_1$ for all $\phi_2$ values. In the $t_1 > t_2$ case, the current initially increases with $\lambda_1$ for $\phi_2 = 0$ but decreases at larger $\lambda_1$ values, and overall remains lower; for the other two phases, the current decreases consistently with $\lambda_1$. In the $t_1 < t_2$ scenario, the current for $\phi_2 = \pi/2$ shows a non-monotonic behavior, first increasing and then decreasing with $\lambda_1$, although the overall magnitude is smaller than in the $t_1 > t_2$ case. For quarter filling, irrespective of the hopping correlation or phase configuration, the current decreases monotonically with $\lambda_1$. These observations highlight why $\lambda_2$ is chosen as the key tuning parameter in our study: varying $\lambda_2$ allows access to multiple peaks in current, regions of stabilization where the current remains nearly constant, and overall higher current values for both half- and quarter-filling cases, providing a richer control over charge transport in the system.

---

[†] Electronic address: souvikroy138@gmail.com

[‡] Electronic address: ranjinibhattacharya@gmail.com

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
