# Peer review of "Tunable Quantum Transport in Flux-Driven Designer Rings: Role of Hopping Dimerization, Electron Filling, and Phase Architecture"

_SciPost Physics Core_

## Round 1 · Referee Report · Anonymous (Referee 1) · 2025-8-20

Report
Recommendation
Ask for major revision
We apologize for the confusion and thank the referee for the constructive suggestions. In
response, we have made the following clarifications and additions:
• We have included additional references related to the generalized AAH model.
• In the Introduction, Subsection K, and Appendix D, we now provide a clearer justification and
motivation for the particular model considered in our work
(In rings with random or Aubry–Andr´e–Harper (AAH) site-energy modulation, the persistent
current decreases monotonically with disorder and vanishes in the strong limit, with next-
nearest-neighbor hopping offering only minor enhancement. In contrast, nearest-neighbor hop-
ping with unit-cell modulation exhibits a nonmonotonic response, showing a disorder-induced
peak at half-filling. The generalized AAH (GAAH) model, ϵ_n = λ cos(2πbn+ϕ)/[1−α cos(2πbn+
ϕ)], introduces an additional tunable parameter α that enables independent control of the mod-
ulation strength, giving rise to mobility edges and richer localization characteristics. Although
extensive studies have examined localization phenomena in GAAH systems, the behavior of
persistent current in such settings remains largely unexplored. Motivated by this gap, we inves-
tigate transport within a non-interacting tight-binding framework on the Su–Schrieffer–Heeger
(SSH) lattice incorporating GAAH modulation in a one-dimensional ring geometry.)
• As the manuscript is already quite lengthy, we have added an Appendix where we discuss
in detail the localization-delocalization aspects, state currents, including system size depen-
dence and, more specifically, the rationale for choosing our model and the essential parameter
variations.
• As the results pertaining to the conventional AAH model are well established in earlier studies,
we have duly cited the corresponding references (Refs. 63–65) to acknowledge the existing body
of work.

Author: Souvik Roy on 2025-10-06 [id 5889]
(in reply to Report 2 on 2025-08-22)RESPONSE TO THE COMMENTS MADE BY THE REVIEWER – 2::
Comment: At this stage, the manuscript is not in the position to be accepted. There is not
enough discussion on the background, AAH model and the connection to the quantities studied.
All the results are mostly numerical observations without any strong physical arguments such as
phenomenological pictures to back them. Some observations are completely left without explana-
tion. For example, 1. in Sec. III J, why the re-entrant transitions and what are the expectations
for this transition? 2. Another point is why tuning phi to such values, what is so special about
choosing phi=pi/2. 3. The connection between average IPR and NPR with the current seems to
be mere accidental unless backed with proper physical picture. Considering all of these factors I
do not recommend this manuscript for publication unless major revision is done to include physical
arguments, enough discussions to back the numerical observations.
Reply: We are extremely sorry about the earlier lack of clarity and sincerely thank the referee
for pointing this out. In response, we have carefully revised the manuscript to improve clarity and
organization. The major changes and clarifications are as follows:
• The background discussion, literature survey, and the motivation behind our proposed model
have been elaborated in Subsection K. In addition, the motivation is now presented more clearly
in the Introduction section.
• The content of Section III, Subsection H has been comprehensively rewritten for better orga-
nization, and Subsection J has been revised with a new paragraph to improve readability and
to present the physical insights more transparently.
• The justification for choosing specific values of ϕ2 (such as 0, π, and π/2) is now explicitly
provided in Subsection K
(In this model, we introduce two Aubry–Andr´e–Harper (AAH) phases, ϕ1 and ϕ2, and examine
three representative cases of ϕ2 while fixing ϕ1 = 0, namely ϕ2 = 0, π, and π/2, corresponding
to distinct physical configurations. For ϕ2 = 0, both sites in a unit cell share identical poten-
tials (1 + λ2 cos 2πib), yielding a non-staggered profile. When ϕ2 = π, the potentials become
(1 + λ2 cos 2πib) and (1 − λ2 cos 2πib), forming a staggered pattern. For ϕ2 = π/2, the sites
acquire (1 + λ2 cos 2πib) and (1 − λ2 sin 2πib), producing a mixed cosine–sine landscape. These
three phase choices enable a systematic exploration of how non-staggered, staggered, and hybrid
modulations govern localization, delocalization, and transport characteristics.).
• The connection between the inverse participation ratio (IPR), normalized participation ratio
(NPR), and the charge current has been clearly established. In particular, we present the
current-NPR relation using both the state current-NPR framework and the NPR-eigenstate
framework in Appendix B. We have moved this detailed discussion to the appendix to avoid
further lengthening the main text.
Attachment:
Reply2.pdf

---

## Round 1 · Referee Report · Anonymous (Referee 2) · 2025-8-22

Strengths
Weaknesses
- Lack of proper motivation for studying this particular model. Not enough background discussed.
- Too many results but poorly threaded together.
- Lack of arguments to back the numerical observations.
Report
Considering all of these factors I do not recommend this manuscript for publication unless major revision is done to include physical arguments, enough discussions to back the numerical observations.
Recommendation
Ask for major revision

---

## Round 2 · Referee Report · Anonymous (Referee 1) · 2025-10-23

Report

The authors made sufficient efforts to meet the acceptance criteria and answer to my questions clarifying the motivation for the choice of the model.

Recommendation

Publish (meets expectations and criteria for this Journal)

---

## Round 2 · Referee Report · Anonymous (Referee 2) · 2025-10-27

Report

The authors have done a significant job in making the recommended revision. The revised manuscript has enough arguments complimenting the numerical evidences, including the appendix. The motivation, results (though mostly numerical still) are clearly established. It can be published.

Recommendation

Publish (meets expectations and criteria for this Journal)

---

## Round 2 · Author Response

Response to the comments made by the reviewers for the manuscript
(Ref: Manuscript ID -scipost 202507 00070v1)

We are very much thankful to both the reviewers as well as the editor for giving us the opportunity
to revise the manuscript and submit it for further consideration.

Following the valuable comments and suggestions of the reviewers, we have amended the manuscript.
In the revised manuscript, all the changes have been highlighted in red color.

Our detailed response to all the comments of the reviewers is as follows.

RESPONSE TO THE COMMENTS MADE BY THE REVIEWER – 1 ::

Comment: I read the article and I do not recommend it for publication in its present form. There
are no references to generalized AAH models, there is not a clear motivation for the choice of that
particular AAH model compared to the conventional one, there is no discussion of the properties
of the unconventional AAH model used (without magnetic flux and SSH), and finally there is no
comparison with the results one would get considering the SSH and the conventional AAH. If there
were a clear discussion about the particular choice of the model, I could deepen the reading but at
the moment, in its present form, it seems more like a numerical exercise.

Reply: We apologize for the confusion and thank the referee for the constructive suggestions. In
response, we have made the following clarifications and additions:

• We have included additional references related to the generalized AAH model.

• In the Introduction, Subsection K, and Appendix D, we now provide a clearer justification and
motivation for the particular model considered in our work

(In rings with random or Aubry–Andr´e–Harper (AAH) site-energy modulation, the persistent
current decreases monotonically with disorder and vanishes in the strong limit, with next-
nearest-neighbor hopping offering only minor enhancement. In contrast, nearest-neighbor hop-
ping with unit-cell modulation exhibits a nonmonotonic response, showing a disorder-induced
peak at half-filling. The generalized AAH (GAAH) model, ϵ_n = λ cos(2πbn+ϕ)/[1−α cos(2πbn+
ϕ)], introduces an additional tunable parameter α that enables independent control of the mod-
ulation strength, giving rise to mobility edges and richer localization characteristics. Although
extensive studies have examined localization phenomena in GAAH systems, the behavior of
persistent current in such settings remains largely unexplored. Motivated by this gap, we inves-
tigate transport within a non-interacting tight-binding framework on the Su–Schrieffer–Heeger
(SSH) lattice incorporating GAAH modulation in a one-dimensional ring geometry.)

• As the manuscript is already quite lengthy, we have added an Appendix where we discuss
in detail the localization-delocalization aspects, state currents, including system size depen-
dence and, more specifically, the rationale for choosing our model and the essential parameter
variations.

• As the results pertaining to the conventional AAH model are well established in earlier studies,
we have duly cited the corresponding references (Refs. 63–65) to acknowledge the existing body
of work.

RESPONSE TO THE COMMENTS MADE BY THE REVIEWER – 2::

Comment: At this stage, the manuscript is not in the position to be accepted. There is not
enough discussion on the background, AAH model and the connection to the quantities studied.
All the results are mostly numerical observations without any strong physical arguments such as
phenomenological pictures to back them. Some observations are completely left without explana-
tion. For example, 1. in Sec. III J, why the re-entrant transitions and what are the expectations
for this transition? 2. Another point is why tuning phi to such values, what is so special about
choosing phi=pi/2. 3. The connection between average IPR and NPR with the current seems to
be mere accidental unless backed with proper physical picture. Considering all of these factors I
do not recommend this manuscript for publication unless major revision is done to include physical
arguments, enough discussions to back the numerical observations.

Reply: We are extremely sorry about the earlier lack of clarity and sincerely thank the referee
for pointing this out. In response, we have carefully revised the manuscript to improve clarity and
organization. The major changes and clarifications are as follows:

• The background discussion, literature survey, and the motivation behind our proposed model
have been elaborated in Subsection K. In addition, the motivation is now presented more clearly
in the Introduction section.

• The content of Section III, Subsection H has been comprehensively rewritten for better orga-
nization, and Subsection J has been revised with a new paragraph to improve readability and
to present the physical insights more transparently.

• The justification for choosing specific values of ϕ2 (such as 0, π, and π/2) is now explicitly
provided in Subsection K

(In this model, we introduce two Aubry–Andr´e–Harper (AAH) phases, ϕ1 and ϕ2, and examine
three representative cases of ϕ2 while fixing ϕ1 = 0, namely ϕ2 = 0, π, and π/2, corresponding
to distinct physical configurations. For ϕ2 = 0, both sites in a unit cell share identical poten-
tials (1 + λ2 cos 2πib), yielding a non-staggered profile. When ϕ2 = π, the potentials become
(1 + λ2 cos 2πib) and (1 − λ2 cos 2πib), forming a staggered pattern. For ϕ2 = π/2, the sites
acquire (1 + λ2 cos 2πib) and (1 − λ2 sin 2πib), producing a mixed cosine–sine landscape. These
three phase choices enable a systematic exploration of how non-staggered, staggered, and hybrid
modulations govern localization, delocalization, and transport characteristics.).

• The connection between the inverse participation ratio (IPR), normalized participation ratio
(NPR), and the charge current has been clearly established. In particular, we present the
current-NPR relation using both the state current-NPR framework and the NPR-eigenstate
framework in Appendix B. We have moved this detailed discussion to the appendix to avoid
further lengthening the main text.

---

## Round 2 · List of Changes

List of changes made

Below we provide the list of changes made in the revised version of our manuscript. All the changes have been highlighted in red color.

  1. The entire manuscript has been thoroughly revised and checked for consistency.

  2. Several new texts, paragraphs, captions, and sub-sections have been incorporated or modified to improve clarity and completeness.

  3. The theoretical description of the individual state current has been added in Sec. B.

  4. A new Subsection K has been added to extend the discussion and include additional theoretical insights.

  5. A total of thirteen figures (Figs. 2–11 and 13–15) have been re-plotted to enhance their visibility and overall presentation quality.

  6. Five new figures have been introduced: Fig. 12 in the main text, and Figs. 16–19 in the Appendix.

  7. Twelve new references (Refs. 63–74) have been added to strengthen the literature support and contextual background.

  8. Four new appendices (appendix A, B, C, and D) have been included to provide additional theoretical and numerical details.

---

## Editorial Decision

accepted_in_target_journal